# DIVIDE TO ADAPT: MITIGATING CONFIRMATION BIAS FOR DOMAIN ADAPTATION OF BLACK-BOX PREDICTORS

**Jianfei Yang**[1]*, **Xiangyu Peng**[2]*, **Kai Wang**[2,4], **Zheng Zhu**[3], **Jiashi Feng**[4], **Lihua Xie**[1],
**Yang You**[2].
[1]Nanyang Technological University    [2]National University of Singapore
[3]Tsinghua University    [4]ByteDance

## ABSTRACT

Domain Adaptation of Black-box Predictors (DABP) aims to learn a model on an unlabeled target domain supervised by a black-box predictor trained on a source domain. It does not require access to both the source-domain data and the predictor parameters, thus addressing the data privacy and portability issues of standard domain adaptation methods. Existing DABP approaches mostly rely on knowledge distillation (KD) from the black-box predictor, *i.e.*, training the model with its noisy target-domain predictions, which however inevitably introduces the confirmation bias accumulated from the prediction noises and leads to degrading performance. To mitigate such bias, we propose a new strategy, *divide-to-adapt*, that purifies cross-domain knowledge distillation by proper domain division. This is inspired by an observation we make for the first time in domain adaptation: the target domain usually contains easy-to-adapt and hard-to-adapt samples that have different levels of domain discrepancy w.r.t. the source domain, and *deep models tend to fit easy-to-adapt samples first*. Leveraging easy-to-adapt samples with less noise can help KD alleviate the negative effect of prediction noises from black-box predictors. In this sense, the target domain can be divided into an easy-to-adapt subdomain with less noise and a hard-to-adapt subdomain at the early stage of training. Then the adaptation is achieved by semi-supervised learning. We further reduce distribution discrepancy between subdomains and develop weak-strong augmentation strategy to filter the predictor errors progressively. As such, our method is a simple yet effective solution to reduce error accumulation in cross-domain knowledge distillation for DABP. Moreover, we prove that the target error of DABP is bounded by the noise ratio of two subdomains, *i.e.,* the confirmation bias, which provides the theoretical justifications for our method. Extensive experiments demonstrate our method achieves state of the art on all DABP benchmarks, outperforming the existing best approach by 9.5% on VisDA-17, and is even comparable with the standard domain adaptation methods that use the source-domain data[1].

## 1 INTRODUCTION

Unsupervised domain adaptation (UDA) (Pan & Yang, 2009) aims to transfer knowledge from a labeled source domain to an unlabeled target domain and has wide applications (Tzeng et al., 2015; Hoffman et al., 2018; Zou et al., 2021). However, UDA methods require to access the source-domain data, thus raising concerns about data privacy and portability issues. To solve them, Domain Adaptation of Black-box Predictors (DABP) (Liang et al., 2022) was introduced recently, which aims to learn a model with only the unlabeled target-domain data and a black-box predictor trained on the source domain, *e.g.*, an API in the cloud, to avoid the privacy and safety issues from the leakage of data and model parameters.

---

*Equal contribution (yang0478@ntu.edu.sg, xiangyupeng@comp.nus.edu.sg).
[1]Codes are available at `https://github.com/xyupeng/BETA`

A few efforts have been made to solve the DABP problem. One of them is to leverage knowledge distillation (Hinton et al., 2015) and train the target model to imitate predictions from the source predictor (Liang et al., 2022). Another one is to adopt learning with noisy labels (LNL) methods to select the clean samples from the noisy target-domain predictions for model training (Zhang et al., 2021). Though inspiring, they have the following limitations. (i) Learning the noisy pseudo labels for knowledge distillation inevitably leads to confirmation bias (Tarvainen & Valpola, 2017), *i.e.*, accumulated model prediction errors. (ii) The LNL-based methods aims to select a clean subset of the target domain to train the model, which would limit the model's performance due to a decreased amount of usable data for model training. (iii) Existing DABP methods lack theoretical justifications.

To address the aforementioned issues, this work proposes a simple yet effective strategy, divide-to-adapt, which suppresses the confirmation bias by purifying cross-domain knowledge distillation. Intuitively, the divide-to-adapt strategy divides the target domain into an easy-to-adapt subdomain with less prediction noise and a hard-to-adapt subdomain. This is inspired by a popular observation: deep models tend to learn clean samples faster than noisy samples (Arpit et al., 2017). For domain adaptation, we make a similar discovery: deep models tend to learn easy-to-adapt samples faster than hard-to-adapt samples, and thus we can leverage the loss distribution of cross-domain knowledge distillation at the early stage for subdomain division. By taking the easy-to-adapt subdomain as a labeled set and the hard-to-adapt subdomain as an unlabeled set, we can solve DABP problem via leveraging prevailing semi-supervised learning methods (Berthelot et al., 2019; Sohn et al., 2020). The divide-to-adapt strategy purifies the target domain progressively for knowledge distillation while fully utilizing all the target dataset without wasting any samples.

To implement the above strategy, this paper proposes **B**lack-Box Mod**E**l Adap**T**ation by Dom**A**in Division (**BETA**) that introduces two key modules to suppress the confirmation bias progressively. Firstly, we divides the target domain into an easy-to-adapt and hard-to-adapt subdomains by fitting the loss distribution into a Gaussian Mixture Model (GMM) and setting a threshold. The easy-to-adapt samples with less noise help purify the cross-domain knowledge distillation for DABP. Secondly, we propose mutually-distilled twin networks with weak-strong augmentation on two subdomains to progressively mitigate error accumulation. The distribution discrepancy between two subdomains is further aligned by an adversarial regularizer to enable the prediction consistency on the target domain. A domain adaptation theory is further derived to provide justifications for BETA.

We make the following contributions. (i) We propose a novel BETA framework for the DABP problem that iteratively suppresses the error accumulation of model adaptation from the black-box source-domain predictor. To the best of our knowledge, this is the first work that addresses the confirmation bias for DABP. (ii) We theoretically show that the error of the target domain is bounded by the noise ratio of the hard-to-adapt subdomain, and empirically show that this error can be suppressed progressively by BETA. (iii) Extensive experiments demonstrate that our proposed BETA achieves state-of-the-art performance consistently on all benchmarks. It outperforms the existing best method by 9.5% on the challenging VisDA-17 and 2.0% on DomainNet.

## 2  RELATED WORK

**Unsupervised Domain Adaptation.**  Unsupervised domain adaptation aims to adapt a model from a labeled source domain to an unlabeled target domain. Early UDA methods rely on feature projection (Pan et al., 2010a) and sample selection (Sugiyama et al., 2007) for classic machine learning models. With the development of deep representation learning, deep domain adaptation methods yield surprising performances in challenging UDA scenarios. Inspired by two-sample test, discrepancy minimization of feature distributions (Koniusz et al., 2017; Yang et al., 2021b; Xu et al., 2022a) is proposed to learn domain-invariant features (Cui et al., 2020a) based on statistic moment matching (Tzeng et al., 2014; Sun & Saenko, 2016). Domain adversarial learning further employs a domain discriminator to achieve the same goal (Ganin et al., 2016; Zou et al., 2019; Yang et al., 2020b) and achieves remarkable results. Other effective techniques for UDA include entropy minimization (Grandvalet & Bengio, 2005; Xu et al., 2021), contrastive learning (Kang et al., 2019), domain normalization (Wang et al., 2019; Chang et al., 2019), semantic alignment (Xie et al., 2018; Yang et al., 2021a), meta-learning (Liu et al., 2020), self-supervision (Saito et al., 2020), semi-supervsed learning (Berthelot et al., 2021) curriculum learning (Zhang et al., 2017; Shu et al., 2019), intra-domain alignment (Pan et al., 2020), knowledge distillation (Yang et al., 2020a) and

self-training (Chen et al., 2020; Zou et al., 2018). Despite their effectiveness, they require access to the source domain data and therefore invoke privacy and portability concerns.

**Unsupervised Model Adaptation and DABP.** Without accessing the source domain, unsupervised model adaptation, *i.e.*, source-free UDA, has attracted increasing attention since it loosens the assumption and benefits more practical scenarios (Guan & Liu, 2021). Early research provides a theoretical analysis of hypothesis transfer learning (Kuzborskij & Orabona, 2013), which motivates the existence of deep domain adaptation without source data (Liang et al., 2020a; Huang et al., 2021; Li et al., 2020; Xu et al., 2022b). Liang *et al.* propose to train the feature extractor by self-supervised learning and mutual information maximization with the classifier frozen (Liang et al., 2020a). This paper deals with a more challenging problem: *only leveraging the labels from the model trained in the source domain for model adaptation*. Few works have been conducted in this field. (Zhang et al., 2021) proposes a noisy label learning method by sample selection, while (Liang et al., 2022) uses knowledge distillation with information maximization. Whereas, we propose to perform domain division and suppress confirmation bias for cross-domain knowledge distillation.

**Confirmation Bias in Semi-Supervised Learning.** Confirmation bias refers to the noise accumulation when the model is trained using incorrect predictions for semi-supervised or unsupervised learning (Tarvainen & Valpola, 2017). Such bias can cause the model to overfit the noisy feature space and then resist new changes (Arazo et al., 2020). In UDA, pseudo-labeling (Saito et al., 2017; Gu et al., 2020; Morerio et al., 2020) and knowledge distillation (Liang et al., 2020a; Kundu et al., 2019; Zhou et al., 2020) are effective techniques but can be degraded due to confirmation bias. Especially for the transfer task with a distant domain, the pseudo labels for the target domain are very noisy and deteriorate the subsequent epochs of training. To alleviate the confirmation bias, several solutions are proposed including co-training (Qiao et al., 2018; Li et al., 2019), Mixup (Zhang et al., 2018; Chen et al., 2019), and data-augmented unlabeled examples (Cubuk et al., 2019). Our paper proposes BETA which is the first work that formulates and addresses the confirmation bias for DABP.

## 3 METHODOLOGY

The idea of our proposed BETA is to mitigate the confirmation bias for DABP by dividing the target domain into two subdomains with different adaptation difficulties. As shown in Figure 1, BETA relies on two designs to suppress error accumulation, including a domain division module that purifies the target domain into a cleaner subdomain and transfers DABP to a semi-supervised learning task, and a two-networks mechanism (*i.e.*, mutually-distilled twin networks) that further diminishes the self-training errors by information exchange. We firstly introduce the problem formulation and the key modules, and then make the algorithmic instantiation with more details.

### 3.1 PROBLEM FORMULATION

For domain adaptation of black-box predictors, the model has access to a black-box predictor $h_s$ trained by a source domain $\{(\mathbf{x}_i^s, y_i^s)\}_{i=1}^{N_s}$ with $N_s$ labeled samples where $\mathbf{x}_i^s \in \mathcal{X}_s, y_i^s \in \mathcal{Y}_s$, and an unlabeled target domain $\{\mathbf{x}_i^t\}_{i=1}^{N_t}$ with $N_t$ unlabeled samples where $\mathbf{x}_i^t \in \mathcal{X}_t$. Assume that the label spaces are the same across two domains, *i.e.*, $\mathcal{Y}_s = \mathcal{Y}_t$, while the inputs data have different distributions, i.e. $\mathbf{x}_i^s \sim \mathcal{D}_S$ and $\mathbf{x}_i^t \sim \mathcal{D}_T$. In other words, there exists a *domain shift* (Ben-David et al., 2007) between $\mathcal{D}_S$ and $\mathcal{D}_T$. The objective is to learn a mapping model $\mathcal{X}_t \to \mathcal{Y}_t$. Different from standard UDA (Pan et al., 2010b; Tzeng et al., 2014; Long et al., 2017), DABP prohibits the model from accessing the source-domain data $\mathcal{X}_s, \mathcal{Y}_s$, and the parameters of the source model $h_s$. Only a black-box predictor trained on the source domain, *i.e.*, an API, is available. Confront these constraints, we can only resort to the hard predictions of the target domain from the source predictor, *i.e.*, $\tilde{\mathcal{Y}}_t = h_s(\mathcal{X}_t)$, in the DABP setting.

### 3.2 DOMAIN DIVISION

Different from the strategy of directly utilizing $\mathcal{X}_t, \tilde{\mathcal{Y}}_t$ for knowledge distillation (Liang et al., 2022), we propose to divide the target domain $\mathcal{X}_t$ into an *easy-to-adapt* subdomain $\mathcal{X}_t^e \sim \mathcal{D}_e$ and a *hard-to-adapt* subdomain $\mathcal{X}_t^h \sim \mathcal{D}_h$, with $\mathcal{X}_t = \mathcal{X}_e \cup \mathcal{X}_h$. Previous studies show that deep models are prone to fitting clean examples faster than noisy examples (Arpit et al., 2017). In domain adaptation,

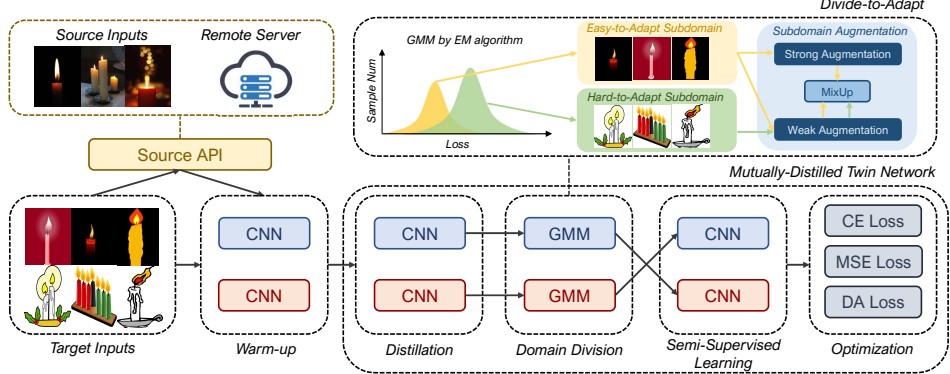

Figure 1: The mutually-distilled twin networks in BETA are initialized by the predictions from the source API. Then the divide-to-adapt strategy is applied for domain division, and the two subdomains with augmentation are leveraged for semi-supervised learning (MSE and CE loss) and domain adaptation (DA Loss, *i.e.*, $\mathcal{L}_{adv}$ and $\mathcal{L}_{mi}$).

the target domain consists of samples that have different similarities to the source-domain samples, and we find that deep models are prone to fitting easy-to-adapt samples first that are more similar to the source domain. Based on the observation, we can obtain the two subdomains with different domain discrepancy by training loss. For example in Figure 2, it is seen that two peaks appear in the loss distribution on the target domain data, and each peak corresponds to one subdomain. We further calculate the A-distance (Ben-David et al., 2010) ($d_A$) between the two subdomains and the source domain, and the result shows that $\mathcal{D}_e$ has less domain discrepancy than $\mathcal{D}_h$. This can be observed more intuitive by the subdomain samples in the appendix.

Inspired by this observation, we first warm up the network, *e.g.*, a CNN, for several epochs, and then obtain the loss distribution by calculating the per-sample cross-entropy loss for a $K$-way classification problem as

$$\mathcal{L}_{ce}(\mathbf{x}_i^t) = -\sum_{k=1}^{K} \tilde{y}_i^k \log(h_t^k(\mathbf{x}_i^t)), \quad (1)$$

where $h_t^k$ is the softmax probability for class $k$ from the target model. As shown in Figure 1, the loss distribution appears to be bimodal and two peaks indicate the clean and noisy clusters, which can be fitted by a GMM using Expectation Maximization (EM) (Li et al., 2019). In noisy label learning, the clean and noisy subset division is achieved by a Beta Mixture

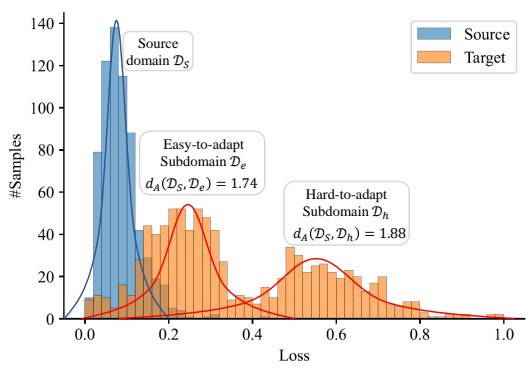

Figure 2: The loss distribution on A→W (Office-31).

Model (BMM) (Arazo et al., 2019). However, in DABP, the noisy pseudo labels obtained by $h_s$ are dominated by asymmetric noise (Kim et al., 2021), *i.e.*, the noisy samples that do not follow a uniform distribution. In this case, the BMM leads to undesirable flat distributions and cannot work effectively for our task (Song et al., 2022). Asymmetric noise in $\tilde{\mathcal{Y}}_t$ also causes the model to perform confidently and generate near-zero losses, which hinders the domain division of GMM. To better fit the losses of the target domain with asymmetric noise, the negative entropy is used as a regularizer in the warm-up phase, defined as:

$$\mathcal{L}_{ne} = \sum_{k=1}^{K} h_t^k(\mathbf{x}_i^t) \log(h_t^k(\mathbf{x}_i^t)). \quad (2)$$

After fitting the loss distribution to a two-component GMM via the Expectation-Maximization algorithm, the clean probability $\varrho_i^c$ is equivalent to the posterior probability $p(c|l_i(\mathbf{x}_i^t))$ where $c$ is the Gaussian component with smaller loss and $l_i(\mathbf{x}_i^t)$ is the cross-entropy loss of $\mathbf{x}_i^t$. Then the clean and

noisy subdomains are divided by setting a threshold $\tau$ based on the clean probabilities:

$$\mathcal{X}_e = \{(\mathbf{x}_i, \tilde{y}_i) | (\mathbf{x}_i, \tilde{y}_i) \in (\mathcal{X}_t, \tilde{\mathcal{Y}}_t), \varrho_i^c \geq \tau\}, \tag{3}$$

$$\mathcal{X}_h = \{(\mathbf{x}_i, \tilde{p}_i) | \mathbf{x}_i \in \mathcal{X}_t, \varrho_i^c < \tau\}, \tag{4}$$

where $\tilde{p}_i = h_s(\mathbf{x}_i^t)$ is the softmax probabilities. Intuitively, the clean subdomain consists of *easy-to-adapt* samples, while the noisy subdomain consists of *hard-to-adapt* samples. The semi-supervised learning methods (Berthelot et al., 2019) can be directly applied with $\mathcal{X}_e$ used as the labeled set and $\mathcal{X}_h$ as the unlabeled set. Compared to sample selection (Zhang et al., 2021) and single distillation (Liang et al., 2022), domain division enables the utilization of all accessible samples by semi-supervised learning and dilutes the risk of the confirmation bias by leveraging cleaner signals of supervision for model adaptation.

### 3.3 Mutually-distilled Twin Networks with Subdomain Augmentation

The easy-to-adapt subdomain is purified by domain division but still has inevitable wrong labels. Overfitting to these wrong labels enforces the model to generate fallacious low losses for domain division, and hence accumulates the wrong predictions iteratively, which is the confirmation bias. Apart from domain division, we propose Mutually-distilled Twin Networks (MTN) to further mitigate such bias, inspired by the two-networks design in (Qiao et al., 2018; Li et al., 2019) where the confirmation bias of self-training can be diminished by training two networks to decontaminate the noise for each other. Specifically, we employ two identical networks initialized independently, where one network performs semi-supervised learning according to the domain division and pseudo labels of the other network. In this fashion, two networks are trained mutually and receive extra supervision to filter the error.

In BETA, we further revamp this design by subdomain augmentation to increase the divergence of domain division, enabling two networks to obtain sufficiently different supervisions from each other. Suppose that two networks $h_t^{\theta_1}, h_t^{\theta_2}$ where $\theta_1, \theta_2$ are parameters generate two sets of domain division $\{\mathcal{X}_e^1, \mathcal{X}_h^1\}$ and $\{\mathcal{X}_e^2, \mathcal{X}_h^2\}$, respectively. We take $h_t^{\theta_2}$ and $\{\mathcal{X}_e^1, \mathcal{X}_h^1\}$ for example. Two augmentation strategies are tailored: the weak augmentation (*e.g.*, random cropping and flipping), and the strong augmentation (*i.e.*, RandAugment (Cubuk et al., 2020) and AutoAugment (Cubuk et al., 2019). The samples from the easy-to-adapt subdomain are mostly correct, so we augment them using two strategies and obtain their soft pseudo labels by the convex combination of averaging all augmentations and the pseudo label according to the clean probability $\varrho_i^c$. Whereas, the hard-to-adapt subdomain is noisy, so we only apply the weak augmentation to update their pseudo labels but use strong augmentations in the subsequent learning phase. Furthermore, we employ the co-guessing strategy (Li et al., 2019) to refine the pseudo labels for $\mathcal{X}_h$. The refined subdomains are derived as:

$$\widehat{\mathcal{X}}_e^1 = \left\{ (\mathbf{x}_i, \tilde{y}_i') | \tilde{y}_i' = \varrho_i^c \tilde{y}_i + (1 - \varrho_i^c) \frac{1}{M} \sum_{m=1}^{M} [h_t^{\theta_2}(A_{w/s}^m(\mathbf{x}_i))], (\mathbf{x}_i, \tilde{y}_i) \in \mathcal{X}_e^1 \right\}, \tag{5}$$

$$\widehat{\mathcal{X}}_h^1 = \left\{ (\mathbf{x}_i, \tilde{p}_i') | \tilde{p}_i' = \frac{1}{2M} \sum_{m=1}^{2M} [h_t^{\theta_1}(A^m(\mathbf{x}_i)) + h_t^{\theta_2}(A_w^m(\mathbf{x}_i))], (\mathbf{x}_i, \tilde{p}_i) \in \mathcal{X}_h^1 \right\}, \tag{6}$$

where $A_{w/s}^m(\cdot)$ denotes the $m$-th weak and strong augmentation function, and $M$ denotes the total number of augmentation views.

### 3.4 Algorithmic Instantiation

After the warm-up, domain division, and subdomain augmentation, we detail the algorithmic choices of other modules and the learning objectives of our framework.

**Hard knowledge distillation.** For each epoch, we first perform knowledge distillation from the predictions of the source model $h_s$ by the relative entropy, *i.e.*, the Kullback–Leibler divergence

$$\mathcal{L}_{kd} = \mathbb{E}_{\mathbf{x}_t \in \mathcal{X}_t} D(\tilde{y}_t || h_t(\mathbf{x}_t)), \tag{7}$$

where $D_{KL}(\cdot||\cdot)$ denotes the KL-divergence, and the pseudo label $\tilde{y}_t$ is obtained by the EMA prediction of $h_s(\mathbf{x}_t)$. Different from DINE (Liang et al., 2022) that uses source model probabilities, we only leverage the hard pseudo labels that are more ubiquitous for API services.

**Mutual information maximization.** To circumvent the model to show partiality for categories with more samples during knowledge distillation, we maximize the mutual information as

$$\mathcal{L}_{mi} = H(\mathbb{E}_{x \in \mathcal{X}_t} h_t(x)) - \mathbb{E}_{x \in \mathcal{X}_t} H(h_t(x)), \tag{8}$$

where $H(\cdot)$ denotes the information entropy. This loss works jointly with $\mathcal{L}_{kd}$. Besides, after the semi-supervised learning, we use this loss to fine-tune the model to enforce the model to comply with the cluster assumption (Shu et al., 2018; Liang et al., 2022; Grandvalet & Bengio, 2005).

**Domain division enabled semi-supervised learning.** We choose MixMatch (Berthelot et al., 2019) as the semi-supervised learning method since it includes a mix-up procedure (Zhang et al., 2018) that can further diverge the two networks while refraining from overfitting. The mixed sets $\ddot{\mathcal{X}}_e, \ddot{\mathcal{X}}_h$ are obtained by $\ddot{\mathcal{X}}_e = \text{Mixup}(\widehat{\mathcal{X}}_e, \widehat{\mathcal{X}}_e \cup \widehat{\mathcal{X}}_h)$ and $\ddot{\mathcal{X}}_h = \text{Mixup}(\widehat{\mathcal{X}}_h, \widehat{\mathcal{X}}_e \cup \widehat{\mathcal{X}}_h)$. Then the loss functions is written as

$$\mathcal{L}_{dd} = \mathcal{L}_{ce}(\ddot{\mathcal{X}}_e) + \mathcal{L}_{mse}(\ddot{\mathcal{X}}_h) + \mathcal{L}_{reg}, \tag{9}$$

where $\mathcal{L}_{ce}$ denotes the cross-entropy loss, $\mathcal{L}_{mse}$ denotes the mean squared error, and the regularizer $\mathcal{L}_{reg}$ uses a uniform distribution $\pi_k$ to eliminate the effect of class imbalance, written as

$$\mathcal{L}_{reg} = \sum_k \pi_k \log \left( \pi_k \Big/ \frac{1}{|\ddot{\mathcal{X}}_e| + |\ddot{\mathcal{X}}_h|} \sum_{\mathbf{x} \in \ddot{\mathcal{X}}_e + \ddot{\mathcal{X}}_h} h_t(\mathbf{x}) \right). \tag{10}$$

**Subdomain alignment.** We assume that there exists a distribution discrepancy between the easy-to-adapt and the hard-to-adapt subdomains, which leading to the performance gap between them. Regarding this gap, we add an adversarial regularizer by introducing a domain discriminator $\Omega(\cdot)$:

$$\mathcal{L}_{adv} = \mathbb{E}_{\mathbf{x} \in \ddot{\mathcal{X}}_e} \log \left( \Omega(h_t(x)) \right) + \mathbb{E}_{\mathbf{x} \in \ddot{\mathcal{X}}_h} \log \left( 1 - \Omega(h_t(x)) \right). \tag{11}$$

**Overall objectives.** Summarizing all the losses, the overall objectives are formulated as

$$\mathcal{L} = \underbrace{(\mathcal{L}_{kd} - \mathcal{L}_{mi})}_{step\,1} + \underbrace{(\mathcal{L}_{dd} - \gamma \mathcal{L}_{adv})}_{step\,2}, \tag{12}$$

where $\gamma$ is a hyper-parameter that is empirically set to 0.1. In step 1, we perform distillation for two networks independently to form tight clusters by maximizing mutual information, while in step 2, the proposed BETA revises their predictions by mitigating the confirmation bias in a synergistic manner. The domain division is performed between two steps.

## 3.5 THEORETICAL JUSTIFICATIONS

Existing theories on UDA error bound (Ben-David et al., 2007) are based on the source-domain data, so they are not applicable to DABP models (Liang et al., 2022), which hinders understanding of these models. To better explain why BETA contributes to DABP, we derive an error bound based on the existing UDA theories (Ben-David et al., 2010). Let $h$ denote a hypothesis, $y_e, y_h$ and $\hat{y}_e, \hat{y}_h$ denote the ground truth labels and the pseudo labels of $\mathcal{X}_e, \mathcal{X}_h$, respectively. As BETA is trained on a mixture of the two subdomains with pseudo labels, the error of BETA can be formulated as a convex combination of the errors of the easy-to-adapt subdomain and the hard-to-adapt subdomain:

$$\epsilon_\alpha(h) = \alpha \epsilon_e(h, \hat{y}_e) + (1 - \alpha)\epsilon_h(h, \hat{y}_h), \tag{13}$$

where $\alpha$ is the trade-off hyper-parameter, and $\epsilon_e(h, \hat{y}_e), \epsilon_h(h, \hat{y}_h)$ represents the expected error of the two subdomains. We derive an upper bound of how the error $\epsilon_\alpha(h)$ is close to an oracle error of the target domain $\epsilon_t(h, y_t)$ where $y_t$ is the ground truth labels of the target domain.

**Theorem 1** *Let $h$ be a hypothesis in class $\mathcal{H}$. Then*

$$|\epsilon_\alpha(h) - \epsilon_t(h, y_t)| \le \alpha(d_{\mathcal{H}\triangle\mathcal{H}}(\mathcal{D}_e, \mathcal{D}_h) + \lambda + \hat{\lambda}) + \rho_h, \tag{14}$$

*where the ideal risk is the combined error of the ideal joint hypothesis $\lambda = \epsilon_e(h^*) + \epsilon_h(h^*)$, the distribution discrepancy $d_{\mathcal{H}\triangle\mathcal{H}}(\mathcal{D}_e, \mathcal{D}_h) = 2\sup_{h,h' \in \mathcal{H}} |\mathbb{E}_{x \sim \mathcal{D}_e}[h(x) \ne h'(x)] - \mathbb{E}_{x \sim \mathcal{D}_n}[h(x) \ne h'(x)]|$, and $\rho_h$ denotes the pseudo label rate of $\hat{y}_h$. The ideal joint hypothesis is given by $h^* = \arg\min_{h \in \mathcal{H}}(\epsilon_e(h) + \epsilon_h(h))$, deriving the ideal risk $\lambda = \epsilon_e(h^*) + \epsilon_h(h^*)$ and the pseudo risk $\hat{\lambda} = \epsilon_e(h^*, \hat{y}_e) + \epsilon_h(h^*, \hat{y}_h)$.*

Table 1: Accuracies (%) on Office-31 for black-box model adaptation. H. Avg. denotes the average accuracy of the hard tasks whose source-only accuracies are below 65%.

| Method | Publication | DABP | A→D | A→W | D→A | D→W | W→A | W→D | Avg. | H. Avg. |
|---|---|---|---|---|---|---|---|---|---|---|
| ResNet-50 | - | - | 79.9 | 76.6 | 56.4 | 92.8 | 60.9 | 98.5 | 77.5 | 58.7 |
| LNL-OT | ICLR-19 | ✓ | 88.8 | 85.5 | 64.6 | 95.1 | 66.7 | 98.7 | 83.2 | 65.7 |
| LNL-KL | BMVC-21 | ✓ | 89.4 | 86.8 | 65.1 | 94.8 | 67.1 | 98.7 | 83.6 | 66.1 |
| HD-SHOT | TPAMI-21 | ✓ | 86.5 | 83.1 | 66.1 | 95.1 | 68.9 | 98.1 | 83.0 | 67.5 |
| SD-SHOT | TPAMI-21 | ✓ | 89.2 | 83.7 | 67.9 | 95.3 | 71.1 | 97.1 | 84.1 | 69.5 |
| DINE | CVPR-22 | ✓ | 91.6 | 86.8 | 72.2 | **96.2** | 73.3 | 98.6 | 86.4 | 72.8 |
| **BETA** (Ours) | - | ✓ | **93.6** | **88.3** | **76.1** | 95.5 | **76.5** | **99.0** | **88.2** | **76.3** |
| BSP+DANN | ICML-19 | ✗ | 93.0 | 93.3 | 73.6 | 98.2 | 72.6 | 100.0 | 88.5 | 73.1 |
| MDD | ICML-19 | ✗ | 93.5 | 94.5 | 74.6 | 98.4 | 72.2 | 100.0 | 88.9 | 73.4 |
| ATDOC | CVPR-21 | ✗ | 94.4 | 94.3 | 75.6 | 98.9 | 75.2 | 99.6 | 89.7 | 75.4 |

Table 2: Accuracies (%) on Office-Home for black-box model adaptation. (':' denotes 'transfer to')

| Method | DABP | Ar:Cl | Ar:Pr | Ar:Re | Cl:Ar | Cl:Pr | Cl:Re | Pr:Ar | Pr:Cl | Pr:Re | Re:Ar | Re:Cl | Re:Pr | Avg. | H. Avg. |
|---|---|---|---|---|---|---|---|---|---|---|---|---|---|---|---|
| ResNet-50 | - | 44.1 | 66.9 | 74.2 | 54.5 | 63.3 | 66.1 | 52.8 | 41.2 | 73.2 | 66.1 | 46.7 | 77.5 | 60.6 | 50.4 |
| LNL-OT | ✓ | 49.1 | 71.7 | 77.3 | 60.2 | 68.7 | 73.1 | 57.0 | 46.5 | 76.8 | 67.1 | 52.3 | 79.5 | 64.9 | 55.6 |
| LNL-KL | ✓ | 49.0 | 71.5 | 77.1 | 59.0 | 68.7 | 72.9 | 56.4 | 46.9 | 76.6 | 66.2 | 52.3 | 79.1 | 64.6 | 55.4 |
| HD-SHOT | ✓ | 48.6 | 72.8 | 77.0 | 60.7 | 70.0 | 73.2 | 56.6 | 47.0 | 76.7 | 67.5 | 52.6 | 80.2 | 65.3 | 55.9 |
| SD-SHOT | ✓ | 50.1 | 75.0 | 78.8 | 63.2 | 72.9 | 76.4 | 60.0 | 48.0 | 79.4 | 69.2 | 54.2 | 81.6 | 67.4 | 58.1 |
| DivideMix | ✓ | 51.7 | 74.7 | 78.5 | 61.8 | 72.4 | 73.3 | 59.8 | 48.0 | 82.9 | 68.0 | 56.4 | 81.6 | 67.4 | 58.4 |
| DINE | ✓ | 52.2 | 78.4 | 81.3 | 65.3 | 76.6 | 78.7 | 62.7 | 49.6 | 82.2 | 69.8 | 55.8 | 84.2 | 69.7 | 60.4 |
| **BETA** (Ours) | ✓ | **57.2** | **78.5** | **82.1** | **68.0** | **78.6** | **79.7** | **67.5** | **56.0** | **83.0** | **71.9** | **58.9** | **84.2** | **72.1** | **64.4** |
| BSP+CDAN | ✗ | 52.0 | 68.6 | 76.1 | 58.0 | 70.3 | 70.2 | 58.6 | 50.2 | 77.6 | 72.2 | 59.3 | 81.9 | 66.3 | 58.1 |
| MDD | ✗ | 54.9 | 73.7 | 77.8 | 60.0 | 71.4 | 71.8 | 61.2 | 53.6 | 78.1 | 72.5 | 60.2 | 82.3 | 68.1 | 60.2 |
| CST | ✗ | 59.0 | 79.6 | 83.4 | 68.4 | 77.1 | 76.7 | 68.9 | 56.4 | 83.0 | 75.3 | 62.2 | 85.1 | 73.0 | 65.3 |

In the above theorem, the error is bounded by the distribution discrepancy between two subdomains, the noise ratio of $\mathcal{X}_h$, and the risks. The ideal risk $\lambda$ is neglectly small (Ganin et al., 2016), and the pseudo risk $\hat{\lambda}$ is bounded by $\rho_h$ as shown in the appendix. Hence, the subdomain discrepancy and $\rho_h$ dominate the error bound. Empirical results show that $d_{\mathcal{H} \triangle \mathcal{H}}(\mathcal{D}_e, \mathcal{D}_h)$ is usually small for the two subdomains, and $\rho_h$ keeps dropping during training as shown in Figure 3(a), which tightens the upper bound consequently. The proof with detailed analytics is in the appendix.

## 4 EXPERIMENTS

### 4.1 SETUP

**Datasets. Office-31** (Saenko et al., 2010) is the most common benchmark for UDA, which consists of three domains (**A**mazon, **W**ebcam, **D**SLR) in 31 categories. **Office-Home** (Venkateswara et al., 2017) consists of four domains (**Ar**t, **Cl**ipart, **Pr**oduct, **Re**al World) in 65 categories, and the distant domain shifts render it more challenging. **VisDA-17** (Peng et al., 2017) is a large-scale benchmark for synthetic-to-real object recognition, with a source domain with 152k synthetic images and a target domain with 55k real images from Microsoft COCO. **DomainNet** (Peng et al., 2019) is the largest DA dataset containing 345 classes in 6 domains: Clipart (clp), Infograph (inf), Painting (pnt), Quickdraw (qdr), Real (rel), Sketch (skt).

**Implementation details.** We implement our method via PyTorch (Paszke et al., 2019), and report the average accuracies among three runs. To show the capacity of handling the confirmation bias, we further report the average accuracies across hard tasks whose source-only accuracies are below 65% (**H. Avg.**). We employ ResNet-50 for Office-31, Office-Home, and DomainNet, and ResNet-101 for VisDA-17 as the backbones (He et al., 2016), and add a new MLP-based classifier, which is commonly used in existing UDA works (Long et al., 2017; Chen et al., 2019; Liang et al., 2022; Saito et al., 2018). The domain discriminator consists of fully-connected layers (2048-256-2) that perform a binary subdomain classification (Long et al., 2018). The ImageNet pre-trained model is utilized as initialization. The model is optimized by mini-batch SGD with the learning rate of 1e-3 for CNN layers and 1e-2 for the MLP classifier. Following DINE (Liang et al., 2022), we use the

Table 3: Accuracies (%) on VisDA-17 for black-box model adaptation.

| Method | DABP | plane | bcycl | bus | car | horse | knife | mcycle | person | plant | sktbrd | train | truck | Per-class | H. Avg. |
|---|---|---|---|---|---|---|---|---|---|---|---|---|---|---|---|
| ResNet-101 | - | 64.3 | 24.6 | 47.9 | 75.3 | 69.6 | 8.5 | 79.0 | 31.6 | 64.4 | 31.0 | 81.4 | 9.2 | 48.9 | 35.2 |
| LNL-OT | ✓ | 82.6 | 84.1 | 76.2 | 44.8 | 90.8 | 39.1 | 76.7 | 72.0 | 82.6 | 81.2 | 82.7 | 50.6 | 72.0 | 71.1 |
| LNL-KL | ✓ | 82.7 | 83.4 | 76.7 | 44.9 | 90.9 | 38.5 | 78.4 | 71.6 | 82.4 | 80.3 | 82.9 | 50.4 | 71.9 | 70.8 |
| HD-SHOT | ✓ | 75.8 | 85.8 | 78.0 | 43.1 | 92.0 | 41.0 | 79.9 | 78.1 | 84.2 | 86.4 | 81.0 | 65.5 | 74.2 | 74.4 |
| SD-SHOT | ✓ | 79.1 | 85.8 | 77.2 | 43.4 | 91.6 | 41.0 | 80.0 | 78.3 | 84.7 | 86.8 | 81.1 | 65.1 | 74.5 | 74.8 |
| DINE | ✓ | 81.4 | 86.7 | 77.9 | 55.1 | 92.2 | 34.6 | 80.8 | 79.9 | 87.3 | 87.9 | 84.3 | 58.7 | 75.6 | 74.3 |
| **BETA** (Ours) | ✓ | **94.9** | **90.2** | **85.4** | **61.1** | **95.5** | **93.1** | **85.0** | **83.8** | **92.9** | **91.9** | **91.1** | 55.0 | **85.1** | **85.9** |
| SAFN | ✗ | 93.6 | 61.3 | 84.1 | 70.6 | 94.1 | 79.0 | 91.8 | 79.6 | 89.9 | 55.6 | 89.0 | 24.4 | 76.1 | 70.9 |
| CDAN+E | ✗ | 94.3 | 60.8 | 79.9 | 72.7 | 89.5 | 86.8 | 92.4 | 81.4 | 88.9 | 72.9 | 87.6 | 32.8 | 78.3 | 74.7 |
| DTA | ✗ | 93.7 | 82.2 | 85.6 | 83.8 | 93.0 | 81.0 | 90.7 | 82.1 | 95.1 | 78.1 | 86.4 | 32.1 | 81.5 | 78.7 |

Table 4: Accuracies (%) on DomainNet for black-box model adaptation. The row indicates the source domain while the column indicates the target domain.

| ResNet | clp | inf | pnt | qdr | rel | skt | Avg. | DINE | clp | inf | pnt | qdr | rel | skt | Avg. | BETA | clp | inf | pnt | qdr | rel | skt | Avg. |
|---|---|---|---|---|---|---|---|---|---|---|---|---|---|---|---|---|---|---|---|---|---|---|---|
| clp | - | 16.5 | 36.0 | 10.1 | 52.8 | 41.8 | 31.4 | clp | - | 12.1 | 29.6 | 11.1 | 60.4 | 37.3 | 29.4 | clp | - | 13.4 | 41.2 | 13.0 | 61.8 | 41.1 | 34.1 |
| inf | 32.1 | - | 32.0 | 2.7 | 47.4 | 26.4 | 28.1 | inf | 29.5 | - | 37.6 | 3.4 | 53.8 | 26.5 | 30.1 | inf | 34.9 | - | 41.6 | 3.7 | 56.8 | 30.7 | 33.6 |
| pnt | 29.6 | 23.2 | - | 4.9 | 36.7 | 27.8 | 24.4 | pnt | 37.3 | 12.9 | - | 4.2 | 60.5 | 34.7 | 29.9 | pnt | 47.3 | 18.4 | - | 3.2 | 62.5 | 41.9 | 34.7 |
| qdr | 11.2 | 1.1 | 1.9 | - | 4.3 | 7.7 | 5.3 | qdr | 9.4 | 0.7 | 3 | - | 8.3 | 6.6 | 5.6 | qdr | 11.7 | 0.9 | 2.1 | - | 9.1 | 8.1 | 6.4 |
| rel | 48.2 | 19.6 | 47.9 | 4.3 | - | 35.6 | 31.1 | rel | 45.1 | 14.4 | 49.7 | 5.5 | - | 35.0 | 29.9 | rel | 46.5 | 15.8 | 50.9 | 5.6 | - | 37.7 | 31.3 |
| skt | 49.1 | 13.5 | 35.5 | 11.5 | 47.1 | - | 31.3 | skt | 43.3 | 10.0 | 39.3 | 11.6 | 57.2 | - | 32.2 | skt | 47.3 | 12.3 | 42.3 | 14.8 | 59.9 | - | 35.3 |
| Avg. | 34.0 | 14.8 | 30.7 | 6.7 | 37.7 | 27.9 | 25.3 | Avg. | 32.9 | 10.0 | 31.8 | 7.2 | 48.0 | 28.0 | 26.2 | Avg. | 37.5 | 12.2 | 35.6 | 8.1 | 50.0 | 31.9 | 28.2 |

suggested training strategies including the momentum (0.9), batch size (64), and weight decay (1e-3). The number of epochs for warm-up is empirically set to 3, and the training epoch is 50 except 10 for VisDA-17. The hyper-parameters of MixMatch are kept as same as the original paper (Berthelot et al., 2019), attached in the appendix. As two networks of MTN perform similarly, we report the accuracy of the first network.

**Baselines.** For a fair comparison, we follow the protocol and training strategy for the source domain in DINE (Liang et al., 2022), and compare our BETA with state-of-the-art DABP methods. Specifically, LNL-KL (Zhang et al., 2021), LNL-OT (Asano et al., 2019), and DivideMix (Li et al., 2019) are noisy label learning methods. HD-SHOT and SD-SHOT obtain the model using pseudo labels and apply SHOT (Liang et al., 2020a) by self-training and the weighted cross-entropy loss, respectively. We also show state-of-the-art standard UDA methods for comparison, including CDAN (Long et al., 2018), MDD (Zhang et al., 2019), BSP (Chen et al., 2019), CST (Liu et al., 2021), SAFN (Xu et al., 2019), DTA (Lee et al., 2019), ATDOC (Liang et al., 2021), MCC (Jin et al., 2020), BA[3]US (Liang et al., 2020b) and JUMBOT (Fatras et al., 2021).

## 4.2 RESULTS

**Performance comparison**. We show the results on Office-31, Office-Home, VisDA-17, and DomainNet in Table 1, 2, 3, and 4 respectively. The proposed BETA achieves the best performances on all benchmarks. On average, our method outperforms the state-of-the-art methods by 1.8%, 2.4%, 9.5%, and 2.0% on Office-31, Office-Home, VisDA-17, and DomainNet, respectively. The improvement is marginal for Office-31 as it is quite simple. Whereas, for the challenging VisDA-17, the BETA gains a huge improvement of 9.5%, even outperforming standard UDA methods, *e.g.*, CDAN, BSP, SAFN, MDD. This demonstrates that the suppression of confirmation bias by BETA can be as effective as the domain alignment techniques.

**Hard transfer tasks with distant domain shift.** Since our method effectively mitigates the confirmation bias, it works more effectively for the hard tasks with extremely noisy pseudo labels from the source-only model. For the hard tasks with lower than 65% source-only accuracy (*i.e.*, H. Avg.), it is observed that the BETA outperforms the second-best method by 3.5%, 4.0%, and 11.6% on Office-31, Office-Home, and VisDA-17, respectively, which beats the normal UDA methods. For DomainNet, the source-only model produces less than 50% accuracy for all of the transfer tasks, which leads to negative transfer for DINE, *e.g.*, qdr→skt. Whereas, BETA achieves robust improvement for most tasks, outperforming DINE by 2.0% in average. This demonstrates that our method can deal with transfer tasks with distant shifts and BETA can alleviate the negative effect of error accumulation caused by source-only models with poor performance.

Table 5: Ablation studies of learning objectives and MTN on Office-Home.

| $\mathcal{L}_{dd}$ | $\mathcal{L}_{kd}$ | $\mathcal{L}_{mi}$ | $\mathcal{L}_{adv}$ | MTN | Ar→Cl | Cl→Ar | Cl→Pr | Pr→Ar | Pr→Cl | Re→Cl | H. Avg. |
|---|---|---|---|---|---|---|---|---|---|---|---|
| | | | | | 44.1 | 54.5 | 63.3 | 52.8 | 41.2 | 46.7 | 50.4 |
| ✓ | | | | | 55.5 | 65.4 | 76.5 | 64.4 | 50.7 | 58.1 | 61.8 |
| | | ✓ | | | 54.6 | 63.8 | 75.3 | 62.3 | 48.0 | 55.7 | 60.0 |
| ✓ | | | | ✓ | 56.6 | 65.0 | 76.7 | 64.1 | 51.6 | 60.4 | 62.4 |
| ✓ | | ✓ | | ✓ | 55.4 | 67.6 | 78.4 | 65.4 | 54.5 | 58.5 | 63.3 |
| ✓ | ✓ | ✓ | | ✓ | 56.8 | 68.1 | 78.7 | 67.3 | 55.3 | 59.2 | 64.2 |
| ✓ | ✓ | ✓ | ✓ | ✓ | 57.2 | 68.0 | 78.6 | 67.5 | 56.0 | 58.9 | 64.4 |

(a) Confirmation bias

(b) Studies on $\tau$ and MTNs.

Figure 3: Quantitative results on the estimated confirmation bias, and hyper-parameter sensitivity.

## 4.3 ANALYSIS

**Ablation study.** We study the effectiveness of key components in BETA, with results shown in Table 5. It is seen that the semi-supervised loss enabled by domain division significantly improves the source-only model by 11.4%. The mutual twin networks, knowledge distillation, and mutual information contribute to 0.6%, 1.1%, and 0.9% improvements, respectively. As the two subdomains drawn from the target domain are quite similar, the distribution discrepancy is not always effective.

**Confirmation bias.** We study the confirmation bias using the noise ratio of the two subdomains in terms of knowledge distillation (K.D.) and BETA on Office-Home (Ar→Cl) to show the effectiveness of the domain division and MTN. As shown in Figure 3(a), the error rate of K.D. only drops for the first a few epochs and then stops decreasing. Whereas, the error rate of BETA keeps decreasing for about 20 epochs since the confirmation bias is iteratively suppressed. The error gap between K.D. and ours on the hard-to-adapt and easy-to-adapt subdomain reaches around 10% and 3%, respectively, validating that our method reduces the error rate $\rho_h$ and minimizes the target error in Theorem 1.

**Hyper-parameter sensitivity and MTN.** We study the hyper-parameter $\tau$ on Office-Home (Cl→Pr) across three runs. We choose $\tau$ ranging from 0.3 to 0.9, as too small $\tau$ leads to noisy domain division while very large $\tau$ leads to a very small number of samples at the easy-to-adapt subdomain. As shown in Figure 3(b), the accuracies of BETA range from 78.2% to 78.8%, and the best result is achieved at 0.8. For the MTN module, it is observed that the two networks of BETA achieve similar trends over different $\tau$, and one network slightly outperforms another consistently. We further plot the Intersection over Union (IoU) between two easy-to-adapt clean subdomains $\mathcal{X}_e^1, \mathcal{X}_e^2$ generated by domain division, and it decreases as $\tau$ gets greater, which means that a larger $\tau$ leads to more difference of the domain division. The diverged domain division can better mitigate the error accumulation for MTN. Thus, the best result at $\tau = 0.8$ is a reasonable trade-off between the divergence of two domain divisions and the sample number of the easy-to-adapt clean subdomain.

## 5 CONCLUSION

In this work, we propose to suppress confirmation bias for DABP. This is achieved by domain division that purifies the noisy labels in cross-domain knowledge distillation. We further develop mutually-distilled twin networks with subdomain augmentation and alignment to mitigate the error accumulation. Besides, we derive a theorem to show why mitigating confirmation bias helps DABP. Extensive experiments over different backbones and learning setups show that BETA effectively suppresses the noise accumulation and achieves state-of-the-art performance on all benchmarks.

**Acknowledgements.** This work is supported by NTU Presidential Postdoctoral Fellowship, "Adaptive Multimodal Learning for Robust Sensing and Recognition in Smart Cities" project fund, at Nanyang Technological University, Singapore. This research is jointly supported by the National Research Foundation, Singapore under its AI Singapore Programme (AISG Award No: AISG2-PhD-2021-08-008). We thank Google TFRC for supporting us to get access to the Cloud TPUs. This work is jointly supported NUS startup grant, the Singapore MOE Tier-1 grant, and the ByteDance grant.

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

# A  APPENDIX

## A.1  PROOF OF THEOREM 1

We prove the **Theorem 1** which extends the learning theories of domain adaptation (Ben-David et al., 2010) for the black-box domain adaptation and provides theoretical justifications for our method.

Denote $\mathcal{X}_t \sim \mathcal{D}_T$ as the target domain with its sample distribution. $\mathcal{X}_e \sim \mathcal{D}_e$ and $\mathcal{X}_h \sim \mathcal{D}_h$ denote the easy-to-adapt clean subdomain and the hard-to-adapt noisy subdomain with their corresponding sample distributions, respectively. Denote $y_e, y_h$ and $\hat{y}_e, \hat{y}_h$ as the ground truth labels and the pseudo labels of $\mathcal{X}_e, \mathcal{X}_h$, respectively. Let $h$ denote a hypothesis. As our method performs training on a mixture of the clean set and the noisy set with pseudo labels, the error of our method can be formulated as a convex combination of the errors of the clean set and the noisy set:

$$\epsilon_\alpha(h) = \alpha\epsilon_e(h, \hat{y}_e) + (1 - \alpha)\epsilon_h(h, \hat{y}_h), \tag{15}$$

where $\alpha$ is the trade-off hyper-parameter, and $\epsilon_e(h, \hat{y}_e), \epsilon_h(h, \hat{y}_h)$ represents the expected error of the easy-to-adapt clean set $\mathcal{X}_e$ and the hard-to-adapt noisy set $\mathcal{X}_h$, respectively, defined by

$$\epsilon_e(h, \hat{y}_e) = \mathbb{E}_{x \sim \mathcal{D}_e}[|h(x) - \hat{y}_e|] \tag{16}$$
$$\epsilon_h(h, \hat{y}_h) = \mathbb{E}_{x \sim \mathcal{D}_h}[|h(x) - \hat{y}_h|]. \tag{17}$$

We use the shorthand $\epsilon_e(h) = \epsilon_e(h, f_e)$ in the proof.

Then, we derive an upper bound of how the error $\epsilon_\alpha(h)$ is close to an oracle error of the target domain $\epsilon_t(h, y_t)$ where $y_t$ is the ground truth labels of the target domain, which is illustrated in **Theorem 1**:

**Theorem 2** *Let $h$ be a hypothesis in class $\mathcal{H}$. Then*

$$|\epsilon_\alpha(h) - \epsilon_t(h, y_t)| \le \alpha(d_{\mathcal{H}\triangle\mathcal{H}}(\mathcal{D}_e, \mathcal{D}_h) + \lambda + \hat{\lambda}) + \rho_h, \tag{18}$$

*where the ideal risk is the combined error of the ideal joint hypothesis $\lambda = \epsilon_e(h^*) + \epsilon_h(h^*)$, the distribution discrepancy $d_{\mathcal{H}\triangle\mathcal{H}}(\mathcal{D}_e, \mathcal{D}_h) = 2\sup_{h,h'\in\mathcal{H}}|\mathbb{E}_{x\sim\mathcal{D}_e}[h(x) \ne h'(x)] - \mathbb{E}_{x\sim\mathcal{D}_h}[h(x) \ne h'(x)]|$, and $\rho_h$ denote the pseudo label rate of $\hat{y}_h$. The ideal joint hypothesis is given by $h^* = \arg\min_{h\in\mathcal{H}}(\epsilon_e(h) + \epsilon_h(h))$, deriving the ideal risk $\lambda = \epsilon_e(h^*) + \epsilon_h(h^*)$ and the pseudo risk $\hat{\lambda} = \epsilon_e(h^*, \hat{y}_e) + \epsilon_h(h^*, \hat{y}_h)$.*

*Proof:*

$$|\epsilon_\alpha(h) - \epsilon_t(h, y_t)|$$
$$= |\alpha\epsilon_e(h, \hat{y}_e) + (1 - \alpha)\epsilon_h(h, \hat{y}_h) - \alpha\epsilon_e(h, y_e) - (1 - \alpha)\epsilon_h(h, y_h)| \tag{19}$$
$$\le \alpha(|\epsilon_e(h, y_e) - \epsilon_h(h, y_h)| + |\epsilon_e(h, \hat{y}_e) - \epsilon_h(h, \hat{y}_h)|) + |\epsilon_h(h, \hat{y}_h) - \epsilon_h(h, y_h)| \tag{20}$$
$$= \alpha(\epsilon_a + \epsilon_b) + \epsilon_c \tag{21}$$

Then we seek the upper bound of $\epsilon_a, \epsilon_b, \epsilon_c$ by applying the triangle inequality for classification errors (Crammer et al., 2008) as stated in **Lemma 1**.

**Lemma 1** *For any hypotheses $f_1, f_2, f_3$ in class $\mathcal{H}$,*

$$\epsilon(f_1, f_2) \le \epsilon(f_1, f_3) + \epsilon(f_2, f_3). \tag{22}$$

For $\epsilon_a$,

$$\epsilon_a = |\epsilon_e(h, y_e) - \epsilon_h(h, y_h)|$$
$$\le |\epsilon_e(h, y_e) - \epsilon_e(h, h^*)| + |\epsilon_e(h, h^*) - \epsilon_h(h, h^*)| + |\epsilon_h(h, h^*) - \epsilon_h(h, y_h)| \tag{23}$$
$$\le \epsilon_e(h^*) + |\epsilon_e(h, h^*) - \epsilon_h(h, h^*)| + \epsilon_h(h^*) \tag{24}$$
$$\le \frac{1}{2}d_{\mathcal{H}\triangle\mathcal{H}}(\mathcal{D}_e, \mathcal{D}_h) + \lambda \tag{25}$$

For $\epsilon_b$,

$$\epsilon_b = |\epsilon_e(h, \hat{y}_e) - \epsilon_h(h, \hat{y}_h)|$$

$$\leq \epsilon_e(h^*, \hat{y}_e) + |\epsilon_e(h, h^*) - \epsilon_h(h, h^*)| + \epsilon_h(h^*, \hat{y}_h) \tag{26}$$

$$\leq \frac{1}{2}d_{\mathcal{H}\triangle\mathcal{H}}(\mathcal{D}_c, \mathcal{D}_n) + (\epsilon_e(h^*, \hat{y}_e) + \epsilon_h(h^*, \hat{y}_h)) \tag{27}$$

$$\leq \frac{1}{2}d_{\mathcal{H}\triangle\mathcal{H}}(\mathcal{D}_e, \mathcal{D}_h) + \hat{\lambda} \tag{28}$$

$$\tag{29}$$

For $\epsilon_c$,

$$\epsilon_c = |\epsilon_h(h, \hat{y}_h) - \epsilon_h(h, y_h)| \leq |\epsilon_h(\hat{y}_h, y_h)| = \rho_h$$

By summarizing $\epsilon_a, \epsilon_b, \epsilon_c$, we yield the inequality in **Theorem 1**:

$$|\epsilon_\alpha(h) - \epsilon_t(h, y_t)| \tag{30}$$

$$\leq \alpha[(\frac{1}{2}d_{\mathcal{H}\triangle\mathcal{H}}(\mathcal{D}_e, \mathcal{D}_h) + \lambda) + (\frac{1}{2}d_{\mathcal{H}\triangle\mathcal{H}}(\mathcal{D}_e, \mathcal{D}_h) + \hat{\lambda})] + \rho_h \tag{31}$$

$$= \alpha(d_{\mathcal{H}\triangle\mathcal{H}}(\mathcal{D}_e, \mathcal{D}_h) + \lambda + \hat{\lambda}) + \rho_h \tag{32}$$

□

Furthermore, the pseudo risk is bounded by the ideal risk, the pseudo rate of the clean set $\rho_e$ and the noisy set $\rho_h$, derived as follows:

$$\hat{\lambda} = \epsilon_e(h^*, \hat{y}_e) + \epsilon_h(h^*, \hat{y}_h) \tag{33}$$

$$\leq (\epsilon_e(h^*, y_e) + \epsilon_e(y_e, \hat{y}_e)) + (\epsilon_h(h^*, y_h) + \epsilon_h(y_h, \hat{y}_h)) \tag{34}$$

$$= \lambda + \epsilon_e(y_e, \hat{y}_e) + \epsilon_h(y_h, \hat{y}_h) \tag{35}$$

$$= \lambda + \rho_e + \rho_h \tag{36}$$

Given a constant $\lambda$, when the easy-to-adapt subdomain is mostly correct, i.e., $\rho_e \approx 0$, the pseudo risk is bounded by the pseudo rate of the noisy set $\rho_h$.

## A.2 HYPER-PARAMETER SETTINGS

We show the hyper-parameters utilized in our experiments in Table 6, including $\tau$ for domain division, $\alpha$ for MixUp, $\lambda_{mse}$ that controls the weight of $\mathcal{L}_{mse}$ and the sharpening factor $T$. In semi-supervised learning, to prevent the noisy samples to cause error accumulation, we set $\lambda_{mse}$ to be 0. The Mixup follows a Beta distribution with $\alpha = 1.0$. The sharpening factor $T = 0.5$. We use $\tau = 0.8$ for Office-31 and Office-Home. In VisDA-17, since the model may not perform confidently for the large challenging dataset, we set $\tau = 0.5$ to ensure sufficient samples in the easy-to-adapt subdomain.

| Hyper-parameter\Dataset | Office-Home | Office-31 | VisDA-17 |
|---|---|---|---|
| $\tau$ | 0.8 | 0.8 | 0.5 |
| $\alpha$ | 1.0 | 1.0 | 1.0 |
| $\lambda_{mse}$ | 0. | 0. | 0. |
| $T$ | 0.5 | 0.5 | 0.5 |

Table 6: Hyper-parameters for different datasets.

## A.3 CONVERGENCE OF LOSSES

Figure 4 shows the convergence of the losses of BETA during the training procedure. The adversarial loss keeps small since the two subdomains are all drawn from the same domain and thus the distribution divergence between the two subdomains should be small. The mutual information is maximized as shown in the curve of $\mathcal{L}_{mi}$. The semi-supervised loss $\mathcal{L}_{dd}$ fluctuates while decreasing since two networks utilize the subdomains obtained by each other for semi-supervised learning, which decreases error accumulation.

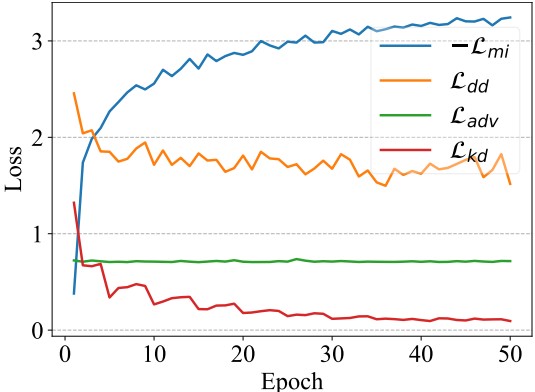

Figure 4: The training procedure of the method.

Table 7: Accuracies (%) on Office-Home for partial-set model adaptation. (':' denotes 'transfer to'.)

| Method | DABP | Ar:Cl | Ar:Pr | Ar:Re | Cl:Ar | Cl:Pr | Cl:Re | Pr:Ar | Pr:Cl | Pr:Re | Re:Ar | Re:Cl | Re:Pr | Avg. | H. Avg. |
|--------|------|-------|-------|-------|-------|-------|-------|-------|-------|-------|-------|-------|-------|------|---------|
| ResNet-50 | - | 44.9 | 70.5 | 80.7 | 57.5 | 61.3 | 67.2 | 60.9 | 40.8 | 76.0 | 70.9 | 47.6 | 76.9 | 62.9 | 52.2 |
| LNL-OT | ✓ | 42.7 | 64.2 | 71.7 | 57.2 | 58.5 | 64.5 | 56.7 | 41.6 | 67.5 | 64.2 | 45.1 | 69.0 | 58.6 | 50.3 |
| LNL-KL | ✓ | 38.9 | 53.8 | 60.5 | 49.2 | 50.5 | 55.9 | 50.0 | 38.9 | 58.0 | 57.0 | 41.7 | 59.6 | 51.2 | 44.9 |
| HD-SHOT | ✓ | 51.2 | 76.2 | 85.7 | 68.8 | 70.6 | 77.5 | 69.2 | 49.6 | 81.4 | 75.9 | 54.1 | 80.7 | 70.1 | 60.6 |
| SD-SHOT | ✓ | 54.2 | 81.8 | 88.9 | 74.8 | 76.5 | 81.0 | 73.5 | 50.6 | 84.2 | 79.8 | 58.4 | 83.7 | 74.0 | 64.7 |
| DINE | ✓ | 58.1 | 83.4 | 89.2 | **78.0** | 80.0 | 80.6 | 74.2 | 56.6 | 85.9 | 80.6 | **62.9** | 84.8 | 76.2 | 68.3 |
| **BETA** (Ours) | ✓ | **61.7** | **88.5** | **91.6** | 77.7 | **80.1** | **86.3** | **75.2** | **58.4** | **87.0** | **81.1** | 61.5 | **86.7** | **78.0** | **69.1** |
| BA³US | ✗ | 60.6 | 83.2 | 88.4 | 71.8 | 72.8 | 83.4 | 75.5 | 61.6 | 86.5 | 79.3 | 62.8 | 86.1 | 76.0 | 67.5 |
| MCC | ✗ | 63.1 | 80.8 | 86.0 | 70.8 | 72.1 | 80.1 | 75.0 | 60.8 | 85.9 | 78.6 | 65.2 | 82.8 | 75.1 | 67.8 |
| JUMBOT | ✗ | 62.7 | 77.5 | 84.4 | 76.0 | 73.3 | 80.5 | 74.7 | 60.8 | 85.1 | 80.2 | 66.5 | 83.9 | 75.5 | 69.0 |

## A.4 APPLICATIONS TO MORE UDA SCENARIOS.

Apart from closed-set UDA, we also demonstrate the effectiveness of our method for partial-set UDA tasks. To this end, we select the first 25 classes in alphabetical order as the target domain from Office-Home. As shown in Table 7, it is seen that LNL-OT and LNL-KL lead to negative transfer due to the label shift. Compared to existing state-of-the-art methods, the proposed BETA achieves the best accuracy of 78.0%, and even outperforms some standard UDA methods (Fatras et al., 2021; Jin et al., 2020). The improvements for partial-set tasks are not large, as BETA is not tailored to address the label shift.

Moreover, the proposed BETA can be easily extended to the semi-supervised domain adaptation and multi-source domain adaptation. For the semi-supervised domain adaptation, we just add the labeled samples in the target domain to the easy-to-adapt subdomain, which enables BETA in a semi-supervised manner (Berthelot et al., 2021). The labeled samples help BETA build a cleaner division for DABP problem. For the multi-source domain adaptation, we can just change the source API to an average prediction or voting of multiple source APIs.

## A.5 EXPERIMENTS UNDER CHALLENGING SCENARIOS.

As the proposed method is partially based on the semi-supervised learning and self-training, there are two factors that might hinder the adaptation capacity of BETA: the number of training samples and the noise ratio. To study if BETA can lead to improvement in the extreme situations, we choose four hard tasks Ar→Cl (44.1%), Cl→Ar (54.5%), Pr→Ar (52.8%), Re→Cl (46.7%), and only choose a super small subset (30 samples per class) of the original domain as the unlabeled target domain data. The results have been shown in Table 8, which demonstrates that our method still brings a large improvement using only a limited number of unlabeled samples with a super low noise ratio. However, the improvement margin is less than that of the original setting (with more samples).

| Method | Ar→Cl | Cl→Ar | Pr→Ar | Re→Cl | Average |
|---|---|---|---|---|---|
| Source-only | 46.97 | 51.65 | 52.00 | 47.28 | 49.48 |
| BETA | 53.79 | 60.32 | 61.10 | 54.26 | 57.37 |

Table 8: Accuracy (%) on Office-Home for challenging situations.

## A.6 ABLATION STUDY ON VISDA-17.

To demonstrate the effectiveness of the proposed design $\mathcal{L}_{dd}$, we further supplement the ablation study on VisDA-17. Note that MTN is applied to all runs in the ablation study. The results are shown in Table 9. It is shown that each loss brings some improvement, but the largest improvement is brought by $L_{dd}$. The combination of $L_{kd}$ and $L_{dd}$ lead to a small decreasing accuracy, due to the very noisy label of the source model that hinders the knowledge distillation. Surprisingly, we find that only our proposed $L_{dd}$ and information maximization can achieve a new state-of-the-art (SOTA) performance of 85.1% on VisDA-17, outperforming existing SOTA method (DINE) by 9.5%. Previously in the manuscript, all the four losses were leveraged for all datasets and experiments. Through this ablation, we can see that the proposed $\mathcal{L}_{dd}$ brings the largest improvement of 36.2% against the source-only model. The performances of BETA can be further improved if we fine-tune the hyper-parameters.

Table 9: Ablation study of four learning objectives on VisDA-17.

| Task | plane | bcycl | bus | car | horse | knife | mcycle | person | plant | sktbrd | train | truck | Per-class |
|---|---|---|---|---|---|---|---|---|---|---|---|---|---|
| Source-only | 64.3 | 24.6 | 47.9 | 75.3 | 69.6 | 8.5 | 79.0 | 31.6 | 64.4 | 31.0 | 81.4 | 9.2 | 48.9 |
| $\mathcal{L}_{kd}$ | 67.9 | 66.2 | 71.4 | 85.9 | 77.6 | 0.0 | 64.4 | 60.8 | 86.1 | 71.4 | 87.7 | 22.9 | 63.5 |
| $\mathcal{L}_{kd} + \mathcal{L}_{mi}$ | 81.4 | 86.7 | 77.9 | 55.1 | 92.2 | 34.6 | 80.8 | 79.9 | 87.3 | 87.9 | 84.3 | 58.7 | 75.6 |
| $\mathcal{L}_{dd} + \mathcal{L}_{mi}$ | 94.9 | 90.2 | 85.4 | 61.1 | 95.5 | 93.1 | 85 | 83.8 | 92.9 | 93.9 | 91.1 | 55 | 85.1 |
| $\mathcal{L}_{kd} + \mathcal{L}_{mi} + \mathcal{L}_{dd}$ | 94.8 | 84.1 | 79.9 | 70.1 | 94.3 | 83.7 | 83.3 | 82.8 | 92.4 | 88.6 | 88.2 | 45.4 | 82.3 |
| $\mathcal{L}_{kd} + \mathcal{L}_{mi} + \mathcal{L}_{dd} + \mathcal{L}_{adv}$ | 96.2 | 83.9 | 82.3 | 71.0 | 95.3 | 73.1 | 88.4 | 80.6 | 95.5 | 90.9 | 88.3 | 45.1 | 82.6 |

## A.7 HYPER-PARAMETER SENSITIVITY ON VISDA-17.

To further validate the hyper-parameter sensitivity, we conduct an additional experiment on VisDA-17. Similarly, we vary $\tau$ from [0.4, 0.7], and the results have been in Table 10. From the results, it is observed that with varying hyper-parameters $\tau$, the proposed method can still achieve significant improvements against the source-only model and the existing state-of-the-art method (DINE). Even the worst case ($\tau$=0.4) brings an improvement of 32.9% against the source-only model, and outperforms the existing state-of-the-art model (DINE) by 6.2%. In real-world applications, we recommend directly using the empirical value (0.6±0.2) that can perform well on all the datasets in this paper. Note that $\tau$ cannot be set to a very large value, as this could lead to a limited number of the easy-to-adapt subdomain. It cannot be set to a very small value, as this could lead to a very noisy split of two subdomains.

Table 10: Sensitivity study of hyper-parameter $\tau$ on VisDA-17.

| Method | plane | bcycl | bus | car | horse | knife | mcycle | person | plant | sktbrd | train | truck | Per-class |
|---|---|---|---|---|---|---|---|---|---|---|---|---|---|
| Source-only | 64.3 | 24.6 | 47.9 | 75.3 | 69.6 | 8.5 | 79.0 | 31.6 | 64.4 | 31.0 | 81.4 | 9.2 | 48.9 |
| DINE | 81.4 | 86.7 | 77.9 | 55.1 | 92.2 | 34.6 | 80.8 | 79.9 | 87.3 | 87.9 | 84.3 | 58.7 | 75.6 |
| BETA ($\tau$=0.4) | 95.1 | 83.0 | 81.8 | 70.4 | 94.5 | 72.6 | 87.7 | 79.4 | 95.0 | 90.5 | 87.6 | 44.5 | 81.8 |
| BETA ($\tau$=0.5) | 96.2 | 83.9 | 82.3 | 71.0 | 95.3 | 73.1 | 88.4 | 80.6 | 95.5 | 90.9 | 88.3 | 45.1 | 82.6 |
| BETA ($\tau$=0.6) | 95.3 | 83.4 | 81.5 | 70.8 | 94.6 | 72.2 | 88.5 | 80.3 | 94.6 | 90.7 | 88.2 | 45.3 | 82.1 |
| BETA ($\tau$=0.7) | 94.3 | 82.3 | 80.7 | 70.1 | 93.8 | 72.1 | 87.1 | 79.9 | 94.2 | 89.8 | 87.2 | 44.3 | 81.3 |

## A.8 COMPARISON TO ADAMATCH, INTRADA, DIVIDEMIX, AND CURRICULUM LEARNING

We compare our method with existing works that may share partial similar ideas. Semi-supervised learning for domain adaptation is proposed in AdaMatch (Berthelot et al., 2021) and IntraDA (Berthelot et al., 2021) proposes to reduce intra-domain discrepancy for semantic segmentation. The differences with these methods lie in the problem formulation, motivation, and the framework design. In this paper, we aim to deal with DABP problem where the model cannot access the source-domain data and model parameters, while these works highly rely on the source data. Without any labeled data, our method is motivated by a new observation, and performs domain division to generate two subdomains. Then we design the twin network structures to further mitigate the confirmation bias during self-training.

We also see some works that proposes easy-to-hard strategy (Cui et al., 2020b; Shin et al., 2020; Shu et al., 2019). However, all of these works require the source domain data for training, and thus these papers cannot be used in our scenario. Besides, these works rely on intermediate domain generation or curriculum learning, none of which leverages our observation and idea, "deep models tend to fit easy-to-adapt samples". We further compare our method with the DivideMix (Li et al., 2019) that draws the similar observation in noisy-label learning.

Table 11: The differences between our method and DivideMix.

|  | BETA | DivideMix |
|---|---|---|
| Task | Domain Adaptation of Black-box Predictior (DABP). the noise of the target domain in DABP is caused by domain shift between two different domains. | Learning with Noisy Labels (LNL). The noise of LNL is randomly added. |
| Method | (a) BETA uses GMM to divide the target domain and Mix-Match with different augmentation for semi-supervised learning. (similar) (b) BETA applies knowledge distillation between models in parallel to the semi-supervised learning, which is purified by the subdomain division to suppress error accumulation during distillation. (c) Subdomain alignment is proposed to align the internal domain shift. (d) Subdomain augmentation is proposed to enhance structural regularization (i.e., mutual information and mix-up). Strong-weak augmentation fully utilizes the high-confidence samples in $X_e$ and single weak augmentation does not introduce more noise to $X_h$. This process enhances the $L_{mi}$ in Eq.(8) that encourages the model to better comply with the cluster assumption and prevent the partiality for categories. | DivideMix uses GMM to divide the data and then use MixMatch for semi-supervised learning. |
| Theory | BETA analyzes the algorithm design theoretically and its connection with the learning shift of DABP. A new bound of DABP is derived to explain the rationale behind the optimization. | N.A. |
| Experiments | Experiments on Office-Home demonstrate that BETA outperforms DivideMix by 4.7% on average. For the hard tasks with distant domain shift, BETA outperforms DivideMix by 6.0% on average. | Experiments are conducted on LNL benchmarks. |

## A.9 STANDARD DEVIATION

For the experimental results in this paper, we run the codes for 3 time using random seeds. Due to the page limit, we only report the mean accuracy in the paper. Here we further provide the standard deviation (std) in Table 12, which shows that our method can achieve a robust improvement in these

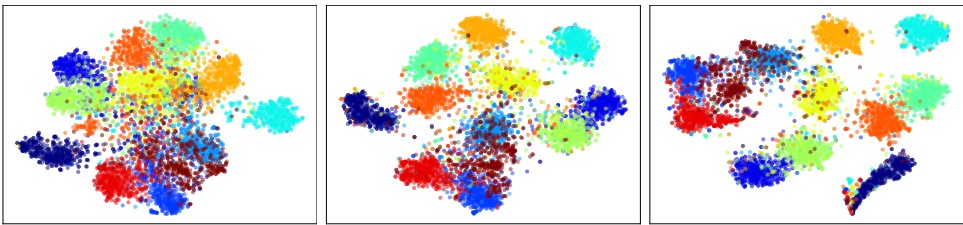

Figure 5: The t-SNE visualization of the target domain on the VisDA-17 dataset at the 1st, 3rd, and 10th training epoch (left to right). Each color indicates one category of VisDA-17.

datasets. We show the std values for all tasks in Office-Home, Office-31 and VisDA-17, and the average std for DomainNet.

Table 12: Standard deviation of the results on all benchmarks.

| Office-31 | | | | | | | | | | | |
|---|---|---|---|---|---|---|---|---|---|---|---|
| | | A→D | A→W | D→A | D→W | W→A | W→D | Avg. | | | |
| | | 0.1 | 0.3 | 0.3 | 0.2 | 0.1 | 0.2 | 0.2 | | | |
| Office-Home | | | | | | | | | | | |
| Ar→Cl | Ar→Pr | Ar→Re | Cl→Ar | Cl→Pr | Cl→Re | Pr→Ar | Pr→Cl | Pr→Re | Re→Ar | Re→Cl | Re→Pr | Avg. |
| 0.3 | 0.5 | 0.2 | 0.3 | 0.3 | 0.2 | 0.3 | 0.1 | 0.5 | 0.2 | 0.3 | 0.3 | 0.3 |
| VisDA-17: 0.5, DomainNet: 0.3 | | | | | | | | | | | |

## A.10 T-SNE VISUALIZATION OF BETA.

Figure 5 shows the feature distribution of the target domain at the 1st, 3rd, and 10th training epoch, and the color indicates the category of VisDA. It is observed that the clusters get tighter with clearer boundaries during training, though there still exists some intrinsic confusion among some classes that remains to be tackled in the future.

## A.11 CODES AND DATASETS

We have attached the codes in the supplementary materials. The *README.md* introduces the two steps: (i) train a source-only model, and (ii) train the BETA using the hard predictions of the source-only model. The datasets should be prepared in the *data* folder using the official websites and their licenses should be followed (Saenko et al., 2010; Venkateswara et al., 2017; Peng et al., 2017).

## A.12 EFFECTIVENESS OF DOMAIN DIVISION

In Figure 6, we show the domain division results at the first epoch (after the warm-up) on Office-Home (Art→Clipart). The three rows contain three categories: alarm clocks, candles, and TV (monitors). The domain shift is very large between *Art* and *Clipart*, and the source-only accuracy is only 44.1%. Even so, the domain division module still accurately divides the clean easy-to-adapt subdomain and the hard-to-adapt subdomain. In the easy-to-adapt subdomain, the contours of objects are similar to those of the source domain, such as the alarm clock. The domain shift between the easy-to-adapt subdomain and the source domain is smaller, as shown in the candle samples with a black background. For the TV, the easy-to-adapt samples have very clear contours and are easy to recognize. In comparison, the hard-to-adapt subdomain is more challenging in terms of shape, color, and style. Our domain division strategy outputs an AUC of 0.814 for the binary classification of clean samples and noisy samples whose pseudo labels are generated by the source-only model, which enables the semi-supervised learning in BETA to be reasonable. During the training, the AUC keeps increasing to 0.828 and further mitigates the confirmation bias progressively.

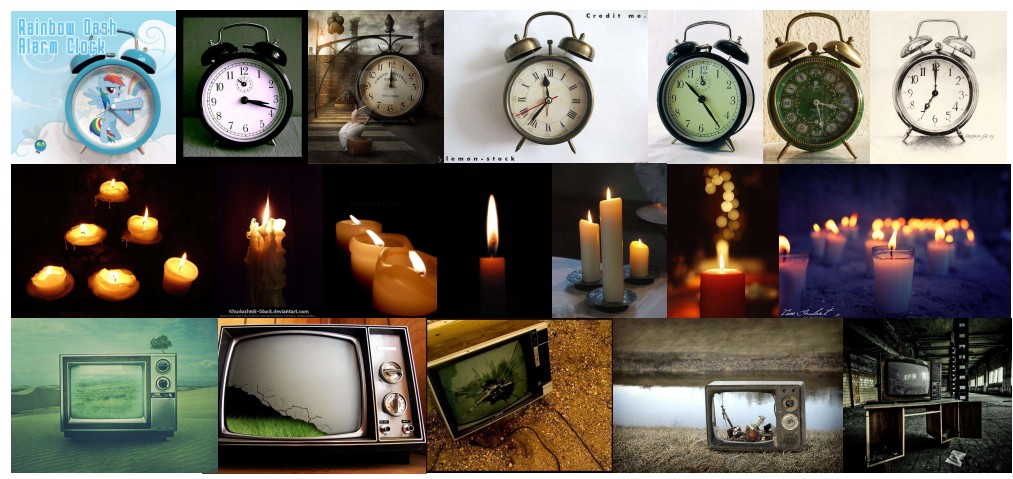

(a) Source domain (Art)

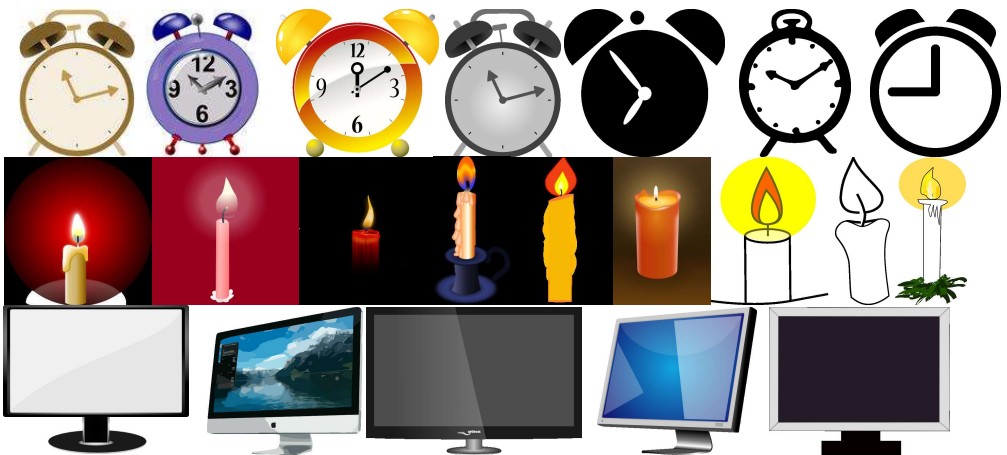

(b) Easy-to-adapt subdomain (Clipart)

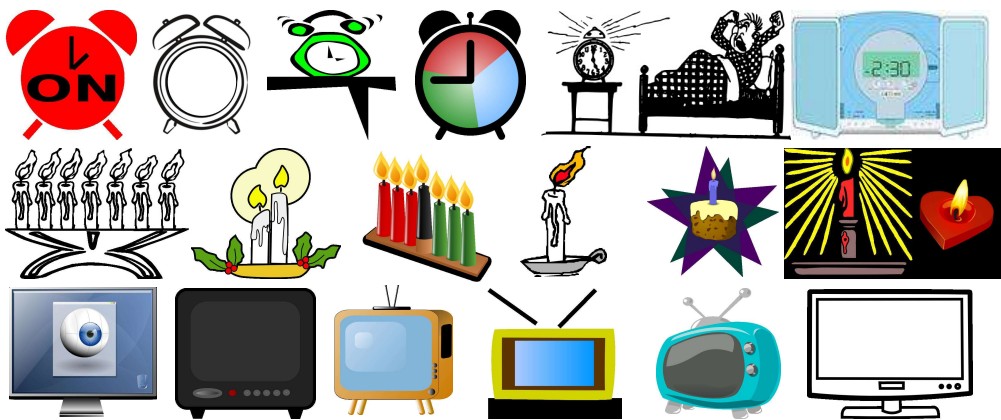

(c) Hard-to-adapt subdomain (Clipart)

Figure 6: The domain division results on Office-Home (Art→Clipart).

