# OpenReview forum: "Divide to Adapt: Mitigating Confirmation Bias for Domain Adaptation of Black-Box Predictors"
_ICLR.cc/2023/Conference — ICLR 2023 notable top 25%_

### Official Review · Reviewer_jpbu · 2022-10-21

**Confidence:** 2
**Correctness:** 3
**Technical Novelty And Significance:** 2
**Empirical Novelty And Significance:** 2
**Recommendation:** 6

**Clarity, Quality, Novelty And Reproducibility:**

The paper is well written but table contents are quite small hopefully I'm reading it from a PDF that I can zoom....

**Strength And Weaknesses:**

Strength:
- the method is well introduced
- experiments are sounding

Weaknesses:
- in the ablation studies more experiment could have been done on the splitting hyper parameter \tau especially on more than one dataset.
this experiment could also look  qualitatively where the splitting occurs in the dataset.

**Summary Of The Paper:**

This work address the Domain Adaptation of Black-box Predictors (DABP) by using a "dive to adapt" strategy.

The contribution are:
- a new divide to adapt strategy
- Theoritical analysis that demonstrates the boundary on the error on the target domain
- experiments to compare with state of the art benchmarks.

**Summary Of The Review:**

This work proposed a simple yet effective framework for DABP.

Few analysis (quantitative and qualitative) are done on the influence of the \tau parameter of the splitting step.
It appears not to be influent on one dataset but that is the easiest dataset...

---

> ### Author Response · Authors · 2022-11-14
> **Response to Reviewer jpbu**
>
> We sincerely thank the reviewer jpbu for the appreciation and constructive comments. We are glad that the reviewer acknowledge our method design and sounding experiments. Here we address the biggest concern raised by the reviewer, i.e., the study on hyper-parameter $\tau$, which determines the viability of BETA in the real world, and hope our response can address this concern.
>
> **Question: In the ablation studies more experiment could have been done on the splitting hyper parameter $\tau$, especially on more than one dataset.**
>
> Thanks for the valuable question. We also realize that the hyper-parameter study is not enough on the Office-Home dataset. To further validate the hyper-parameter sensitivity, we supplement an additional experiment on the large-scale dataset, VisDA-17. Similarly, we evaluate the proposed BETA with the hyper-parameter $\tau$ ranging from [0.4, 0.7], and the results have been shown as follows:
>
> | Method | plane | bcycl | bus | car | horse | knife | mcycle | person | plant | sktbrd | train | truck | Per-class |
> |---|---|---|---|---|---|---|---|---|---|---|---|---|---|
> | Source-only | 64.3 | 24.6 | 47.9 | 75.3 | 69.6 | 8.5 | 79.0 | 31.6 | 64.4 | 31.0 | 81.4 | 9.2 | 48.9 |
> | DINE | 81.4 | 86.7 | 77.9 | 55.1 | 92.2 | 34.6 | 80.8 | 79.9 | 87.3 | 87.9 | 84.3 | 58.7 | 75.6 |
> | BETA ($\tau$=0.4) | 95.1 | 83.0 | 81.8 | 70.4 | 94.5 | 72.6 | 87.7 | 79.4 | 95.0 | 90.5 | 87.6 | 44.5 | 81.8 |
> | BETA ($\tau$=0.5) | 96.2 | 83.9 | 82.3 | 71.0 | 95.3 | 73.1 | 88.4 | 80.6 | 95.5 | 90.9 | 88.3 | 45.1 | 82.6 |
> | BETA ($\tau$=0.6) | 95.3 | 83.4 | 81.5 | 70.8 | 94.6 | 72.2 | 88.5 | 80.3 | 94.6 | 90.7 | 88.2 | 45.3 | 82.1 |
> | BETA ($\tau$=0.7) | 94.3 | 82.3 | 80.7 | 70.1 | 93.8 | 72.1 | 87.1 | 79.9 | 94.2 | 89.8 | 87.2 | 44.3 | 81.3 |
>
> The results show that with varying hyper-parameters $\tau$, the proposed method can still achieve significant improvements against the source-only model and the existing state-of-the-art method (DINE). Even the worst case ($\tau$=0.4) brings an improvement of 32.9% against the source-only model and outperforms the existing state-of-the-art model (DINE) by 6.2%. In real-world applications, we recommend using the empirical value (0.6$\pm$0.2) that can perform well on all the datasets in this paper. Note that $\tau$ cannot be set to a very large value, as this could lead to a limited number of the easy-to-adapt subdomain. It cannot be set to a very small value, which could lead to a very noisy split.
>
> Inspired by this hyper-parameter study, we also want to prove the effectiveness of each part using a large dataset.
>
> | Ablation study | plane | bcycl | bus | car | horse | knife | mcycle | person | plant | sktbrd | train | truck | Per-class |
> |---|---|---|---|---|---|---|---|---|---|---|---|---|---|
> | Source-only | 64.3 | 24.6 | 47.9 | 75.3 | 69.6 | 8.5 | 79.0 | 31.6 | 64.4 | 31.0 | 81.4 | 9.2 | 48.9 |
> | $L_{kd}$ | 67.9 | 66.2 | 71.4 | 85.9 | 77.6 | 0 | 64.4 | 60.8 | 86.1 | 71.4 | 87.7 | 22.9 | 63.5 |
> | $L_{kd}+L_{im}$ | 81.4 | 86.7 | 77.9 | 55.1 | 92.2 | 34.6 | 80.8 | 79.9 | 87.3 | 87.9 | 84.3 | 58.7 | 75.6 |
> | $L_{dd}+L_{im}$ | 94.9 | 90.2 | 85.4 | 61.1 | 95.5 | 93.1 | 85 | 83.8 | 92.9 | 93.9 | 91.1 | 55 | **85.1** |
> | $L_{kd}+L_{im}+L_{dd}$ | 94.8 | 84.1 | 79.9 | 70.1 | 94.3 | 83.7 | 83.3 | 82.8 | 92.4 | 88.6 | 88.2 | 45.4 | 82.3 |
> | $L_{kd}+L_{im}+L_{dd}+L_{adv}$ | 96.2 | 83.9 | 82.3 | 71.0 | 95.3 | 73.1 | 88.4 | 80.6 | 95.5 | 90.9 | 88.3 | 45.1 | 82.6 |
>
> It is shown that each loss brings some improvement, but the largest improvement is brought by $L_{dd}$. Surprisingly, we find that only our proposed $L_{dd}$ and information maximization can achieve a better performance of 85.1% on VisDA-17, outperforming the existing state-of-the-art method (DINE) by 9.5%. The combination of $L_{kd}$ and $L_{dd}$ lead to a small decreasing accuracy, due to the very noisy label of the source model that hinders the knowledge distillation. Previously, four losses were leveraged for all datasets and experiments. Through this ablation, we can see that the proposed $L_{dd}$ brings the largest improvement of 36.2% against the source-only model. The performances of BETA can be further improved if we fine-tune the hyper-parameters.

---

> ### Author Response · Authors · 2022-11-18
> **Looking forward to your reply**
>
> Dear reviewer, we have provided new experiments and discussions according to your valuable suggestions. May I know if you have any other concerns regarding our work? Thanks!

---

### Official Review · Reviewer_uuuU · 2022-10-25

**Confidence:** 4
**Correctness:** 2
**Technical Novelty And Significance:** 2
**Empirical Novelty And Significance:** 2
**Recommendation:** 8

**Clarity, Quality, Novelty And Reproducibility:**

Clarity: The paper is written nicely and is easy to follow.

Quality & Novelty: As discussed above, a bit of novelty is contained in the proposed method. And I would suggest comparing more relevant works to understand the contribution of this paper.

Reproducibility lacks sometimes.

**Strength And Weaknesses:**

Strength:

1. The paper is nicely written and the methodology is well described.

2. Theoretical validation and thorough experiments indicate that the proposed strategy, divide-to-adapt, is a good solution for black-box domain adaptation.

3. It is also impressive that the proposed method outperforms the sota approach by 7.0% on VisDA-17 and even surpasses standard domain adaptation using source data.

Weaknesses:

1. The idea of considering domain adaptation as the semi-supervised setting has been studied in AdaMatch [a] and dividing the target domain into easy and hard subdomains for adaptation has been explored in IntraDA [b].  However, the above relevant works have not been included in this manuscript. The authors should provide more discussion.

2. The technical contribution feels a bit scattered. As the framework can be decomposed into a set of smaller contributions, each improving the performance by a smaller amount. Though authors have provided an ablation study to analyze the individual contribution of each component. However, I find it is quite hard to line the contributions to the motivations.

3. It seems that the proposed method performs better with fewer categories (7.0% gain on VisDA-17 with 12 categories while 1.8% gain on Office-31 with 31 categories and 2.0% gain on DomainNet with 365 categories). Also, I'd rather like to see the ablation studies of each component on VisDA-17 in Table 5.

4. Currently the target application seems to be a bit narrow, as it is only image classification on three benchmarks. I think it would be very interesting to see the applications to other settings e.g., semi-supervised learning, and semi-supervised domain adaptation like AdaMatch [a].

5. As the proposed domain division relies on threshold \tau, it would be better to provide the mean and std of the numerical results under different random seed for each experiment.

Refs:

[a] David Berthelot, Rebecca Roelofs, Kihyuk Sohn, Nicholas Carlini, Alexey Kurakin: AdaMatch: A Unified Approach to Semi-Supervised Learning and Domain Adaptation. ICLR 2022.

[b] Fei Pan, Inkyu Shin, François Rameau, Seokju Lee, In So Kweon: Unsupervised Intra-Domain Adaptation for Semantic Segmentation Through Self-Supervision. CVPR 2020: 3763-3772.


**Summary Of The Paper:**

Based on a simple observation that intra-domain target samples have different levels of domain discrepancy, this paper proposed a new approach, divide-to-adapt, to address the problem of black-box domain adaptation. The main contributions are two parts: dynamically mitigating the confirmation bias for black-box domain adaptation and providing some theoretical justifications. Experiments on image classification benchmarks demonstrate the effectiveness of the proposed approach.

**Summary Of The Review:**

To sum up, I find this paper tackles a practical problem for domain adaptation without source data and model parameters. Yet, it still requires clarification and some solid empirical support before warranting acceptance of this paper.

---

> ### Author Response · Authors · 2022-11-16
> **Response to Reviewer uUUU (2/2)**
>
> **Q4: The ablation studies on VisDA-17?**
>
> We agree with the reviewer, and further have supplemented the ablation study on VisDA-17. $L_{dd}$ means that we use the proposed $L_{dd}$ and MTN, which denote our main contribution.
>
> | Ablation study | plane | bcycl | bus | car | horse | knife | mcycle | person | plant | sktbrd | train | truck | Per-class |
> |---|---|---|---|---|---|---|---|---|---|---|---|---|---|
> | Source-only | 64.3 | 24.6 | 47.9 | 75.3 | 69.6 | 8.5 | 79.0 | 31.6 | 64.4 | 31.0 | 81.4 | 9.2 | 48.9 |
> | $L_{kd}$ | 67.9 | 66.2 | 71.4 | 85.9 | 77.6 | 0 | 64.4 | 60.8 | 86.1 | 71.4 | 87.7 | 22.9 | 63.5 |
> | $L_{kd}+L_{im}$ | 81.4 | 86.7 | 77.9 | 55.1 | 92.2 | 34.6 | 80.8 | 79.9 | 87.3 | 87.9 | 84.3 | 58.7 | 75.6 |
> | $L_{dd}+L_{im}$ | 94.9 | 90.2 | 85.4 | 61.1 | 95.5 | 93.1 | 85 | 83.8 | 92.9 | 93.9 | 91.1 | 55 | **85.1** |
> | $L_{kd}+L_{im}+L_{dd}$ | 94.8 | 84.1 | 79.9 | 70.1 | 94.3 | 83.7 | 83.3 | 82.8 | 92.4 | 88.6 | 88.2 | 45.4 | 82.3 |
> | $L_{kd}+L_{im}+L_{dd}+L_{adv}$ | 96.2 | 83.9 | 82.3 | 71.0 | 95.3 | 73.1 | 88.4 | 80.6 | 95.5 | 90.9 | 88.3 | 45.1 | 82.6 |
>
> It is shown that each component leads to some improvement, but the largest improvement is brought by $L_{dd}$. The combination of $L_{kd}$ and $L_{dd}$ results in a small decreasing accuracy, due to the very noisy label of the source model that hinders the knowledge distillation on VisDA-17. Surprisingly, we find that only our proposed $L_{dd}$ and information maximization ($L_{im}$) can achieve a new state-of-the-art (SOTA) performance of 85.1% on VisDA-17, outperforming existing SOTA method (DINE) by 9.5%. Previously in the manuscript, all four losses were leveraged for all datasets and experiments without fine-tuning the weights of these losses. Through this ablation, we can see that the proposed $L_{dd}$ brings the largest improvement of 36.2% against the source-only model. The performances of BETA can be further improved on these public datasets if we fine-tune the hyper-parameters.
>
>
> **Minor Questions**:
>
> **Q1: Does BETA apply to more domain adaptation scenarios?**
>
> Yes. BETA can be applied to more scenarios. Actually, we have an additional experiment in the appendix to show that the proposed BETA can also be applied to partial domain adaptation (PDA) where the target domain has less classes than the source domain (such as ImageNet). In this case, a label shift exists between two domains, which is very common in the real-world applications. It is shown that our BETA still outperforms existing DABP methods, and even outperforms some standard UDA methods, such as MCC [c] (67.8%, ECCV-20) and JUMBOT [d] (69.0%, ICML-21). This demonstrates that the proposed method can be generalized to more domain adaptation scenarios while still bringing improvement.
>
> | Method | Ar→Cl | Ar→Pr | Ar→Re | Cl→Ar | Cl→Pr | Cl→Re | Pr→Ar | Pr→Cl | Pr→Re | Re→Ar | Re→Cl | Re→Pr | Avg. | H. Avg. |
> |---|---|---|---|---|---|---|---|---|---|---|---|---|---|---|
> | ResNet-50 | 44.9 | 70.5 | 80.7 | 57.5 | 61.3 | 67.2 | 60.9 | 40.8 | 76.0 | 70.9 | 47.6 | 76.9 | 62.9 | 52.2 |
> | NLL-OT | 42.7 | 64.2 | 71.7 | 57.2 | 58.5 | 64.5 | 56.7 | 41.6 | 67.5 | 64.2 | 45.1 | 69.0 | 58.6 | 50.3 |
> | NLL-KL | 38.9 | 53.8 | 60.5 | 49.2 | 50.5 | 55.9 | 50.0 | 38.9 | 58.0 | 57.0 | 41.7 | 59.6 | 51.2 | 44.9 |
> | HD-SHOT | 51.2 | 76.2 | 85.7 | 68.8 | 70.6 | 77.5 | 69.2 | 49.6 | 81.4 | 75.9 | 54.1 | 80.7 | 70.1 | 60.6 |
> | SD-SHOT | 54.2 | 81.8 | 88.9 | 74.8 | 76.5 | 81.0 | 73.5 | 50.6 | 84.2 | 79.8 | 58.4 | 83.7 | 74.0 | 64.7 |
> | DINE | 58.1 | 83.4 | 89.2 | 78.0 | 80.0 | 80.6 | 74.2 | 56.6 | 85.9 | 80.6 | 62.9 | 84.8 | 76.2 | 68.3 |
> | BETA | 61.7 | 88.5 | 91.6 | 77.7 | 80.1 | 86.3 | 75.2 | 58.4 | 87.0 | 81.1 | 61.5 | 86.7 | 78.0 | 69.1 |
>
>
> **Q2: The mean and std of the numerical results under different random seed for each experiment.**
>
> We agree with the reviewer. The current results in our experimental section are the mean accuracy across three runs with different random seeds, and we have added the std on all benchmarks.
>
> **References**
>
> [a] David Berthelot, Rebecca Roelofs, Kihyuk Sohn, Nicholas Carlini, Alexey Kurakin: AdaMatch: A Unified Approach to Semi-Supervised Learning and Domain Adaptation. ICLR 2022.
>
> [b] Fei Pan, Inkyu Shin, François Rameau, Seokju Lee, In So Kweon: Unsupervised Intra-Domain Adaptation for Semantic Segmentation Through Self-Supervision. CVPR 2020
>
> [c] Jin, Y., Wang, X., Long, M., & Wang, J. (2020, August). Minimum class confusion for versatile domain adaptation. ECCV 2020
>
> [d] Fatras, K., Séjourné, T., Flamary, R., & Courty, N. (2021, July). Unbalanced minibatch optimal transport; applications to domain adaptation. ICML 2021

---

> ### Author Response · Authors · 2022-11-16
> **Response to Reviewer uUUU (1/2)**
>
> We sincerely thank the reviewer uuuU for the very constructive comments. We are glad that the reviewer acknowledge our method design and impressive experiments. Here we address the concerns of our paper, and hope our response can address this concern.
>
> **Major Questions**:
>
> **Q1: The author should discuss two two works that share the similar ideas [a][b].**
>
> We appreciate the relevant works advised by the reviewer. We have carefully read these two papers and compared them with our method. There are some similar ideas between our method and these works. AdaMatch [a] also applies the semi-supervised learning framework for domain adaptation, and IntraDA [b] proposes to reduce intra-domain discrepancy. The differences with these methods lie in the problem, motivation, and framework design. In this paper, we deal with DABP problem where the model cannot access the source-domain data and model, while [a][b] highly rely on the source data. Without any labeled data, our method is motivated by a new observation and performs domain division to generate two subdomains. The mutually-distilled twin network and learning objectives are also totally different from [a][b]. We have added the discussion of these novel works in our related work.
>
> **Q2: The technical contribution feels a bit scattered. Does the technical contribution line to the motivations?**
>
> The main technical contribution consists of the subdomain division with a warm-up, and the mutually-distilled networks that suppress the confirmation bias. Other pieces of techniques, such as mutual information and Mixup, are not technical contributions of BETA but are necessary to form a strong UDA baseline. Therefore, we write them in Section 3.4 “Algorithmic Instantiation” using very few sentences.
>
> The technical contribution complies with the motivation. Our method design is inspired by a new observation in UDA: there exist easy-to-adapt and hard-to-adapt subdomains in the target domain and **deep models tend to fit an easy-to-adapt subdomain first**. Based on the observation, the subdomain division is achieved by a simple warm-up for several epochs so that the model can only learn some knowledge from the easy-to-adapt samples. As the model has not fitted the hard-to-adapt samples, this can be reflected in the loss distribution. Then a GMM is used to fit the loss distribution via the EM algorithm to divide the subdomains. Finally, the two subdomains are fed into the mutually-distilled semi-supervised learning. In summary, the method design is based on the new observation so that we can divide two subdomains with less noise. Using a pretty clean subdomain, we can further leverage our mutually-distilled networks to mitigate the confirmation bias.
>
> **Q3: It seems that the proposed method performs better with fewer categories (7.0% gain on VisDA-17 with 12 categories while 1.8% gain on Office-31 with 31 categories and 2.0% gain on DomainNet with 365 categories).**
>
> This question is very interesting and valuable. Please note that the performance gain (7% on VisDA-17 and 1.8% on Office-31) is compared to the state-of-the-art method, DINE, instead of the source-only model. We think that the numerical improvement should attribute to several reasons.
> 1. Firstly, it depends on the task. The Office-31 is a small and easy dataset, and DINE has reached a very high accuracy (86.4%), so the improvement margin is small. For VisDA-17, our method shows a 7% improvement compared to DINE (74.3%).
> 2. Secondly, it depends on the number of samples in the dataset. In Office-31, the largest Amazon domain has 2.5k images, while the target domain in VisDA-17, the target domain has 55k images. *More images help the DABP problem as we have no access to the source-domain data.*
> 3. Thirdly, it depends on the dataset difficulty. DomainNet has 365 categories with a *long-tailed distribution*, which is the hardest dataset. This leads to a high noise ratio for DABP problem, an average noise rate of 25.3%. Initialized by very noisy pseudo labels, our method still brings 2.9% improvement.
>
> In summary, BETA should work better with more unlabeled training data and a low noise ratio. To further explore the extreme situation, we choose four hard tasks Ar$\to$Cl (44.1%), Cl$\to$Ar (54.5%), Pr$\to$Ar (52.8%), Re$\to$Cl (46.7%), and only choose a super small subset (30 samples per class) as the target domain. The results have been shown as follows:
>
> | Method | Ar→Cl | Cl→Ar | Pr→Ar | Re→Cl | Avg |
> |---|---|---|---|---|---|
> | Source-only | 46.97 | 51.65 | 52.00 | 47.28 | 49.48 |
> | BETA | 53.79 | 60.32 | 61.10 | 54.26 | 57.37 |
>
> It is seen that our method still brings a large improvement using only a very limited number of unlabeled samples with a super low noise ratio. However, the improvement margin is less than that of the original setting (with more samples), which indicates that the margin is related to the data size and noise ratio in DABP. We have added these discussions for better understanding.

---

### Official Review · Reviewer_zK5K · 2022-10-28

**Confidence:** 4
**Clarity, Quality, Novelty And Reproducibility:** This work is overall interesting and …
**Correctness:** 3
**Technical Novelty And Significance:** 3
**Empirical Novelty And Significance:** Not applicable
**Recommendation:** 6

**Strength And Weaknesses:**

Strength:
This paper is overall easy to read. The authors describe the task clearly and the work is well organized. The motivation behind this work is clearly presented, i.e., to alleviate the confirmation bias. It is novel to solve DABP problem with semi-supervised learning method. The authors present a comprehensive comparison with previous works. According to the provided result of experiment, the proposed framework outperforms the competing algorithms.

Weakness:
- I am confused that whenever we get a batch of new test data, is it necessary to do extra studies on threshold value to find a suitable value according to different data distributions?
- More discussion to DivideMix should be added. This work shares similar spirits to dividemix and the authors should clarify it.

**Summary Of The Paper:**

In this work, the authors propose a method to increase the accuracy of classification in the task Domain Adaptation of Black-box Predictors. It aims to suppress the confirmation bias efficiently, and extends this work with a learning framework called BETA. Specifically, inspired by the phenomena that deep models tend to learn easy-to-adapt samples faster than hard-to-adapt samples, the authors divide the target domain into an easy-to-adapt subdomain and a hard-to-adapt subdomain. With treating the former as a labeled set and the latter as an unlabeled set, the authors solve DABP via semi-supervised learning methods. Additionally, the authors propose mutually-distilled twin networks with weak-strong augmentation method to mitigate error accumulation.

**Summary Of The Review:**

I hope the authors can address my concerns in Section II.

---

> ### Author Response · Authors · 2022-11-14
> **Response to Reviewer zK5K (2/2)**
>
> **Q2: More discussion should be compared with DivideMix as this work shares similar spirits to DivideMix.**
>
> We agree with the reviewer and therefore make the following comparisons with DivideMix. These discussions will be supplemented to the final manuscript. The main differences between BETA and DivideMix are illustrated from four perspectives:
>
> |  | BETA | DivideMix |
> |---|---|---|
> | Task | Domain Adaptation of Black-box Predictior (DABP). the noise of the target domain in DABP is caused by domain shift between two different domains. | Learning with Noisy Labels (LNL). The noise of LNL is randomly added. |
> | Method | (a) BETA uses GMM to divide the target domain and MixMatch with different augmentation for semi-supervised learning. (similar) (b) BETA applies knowledge distillation between models in parallel to the semi-supervised learning, which is purified by the subdomain division to suppress error accumulation during distillation. (c) Subdomain alignment is proposed to align the internal domain shift. (d) Subdomain augmentation is proposed to enhance structural regularization (i.e., mutual information and mix-up). Strong-weak augmentation fully utilizes the high-confidence samples in $X_e$ and single weak augmentation does not introduce more noise to $X_h$. This process enhances the $L_{mi}$ in Eq.(8) which encourages the model to better comply with the cluster assumption and prevents the partiality for categories. | DivideMix uses GMM to divide the data and then use MixMatch for semi-supervised learning. |
> | Theory | BETA analyzes the algorithm design theoretically and its connection with the learning shift of DABP.  A new bound of DABP is derived to explain the rationale behind the optimization. | N.A. |
> | Experiments | We added an experiment on Office-Home to demonstrate the performance difference. As shown in the table below,  BETA outperforms DivideMix by 4.7% on average. For the hard tasks with distant domain shift, BETA outperforms DivideMix by 6.0% on average.  | Experiments are conducted on LNL benchmarks. |
>
> The comparison on Office-Home is shown as follows, and our method outperforms the DivideMix by 6.0% and the state-of-the-art DINE by 4.0% on average.
>
> | Method    | Ar→Cl | Ar→Pr | Ar→Re | Cl→Ar | Cl→Pr | Cl→Re | Pr→Ar | Pr→Cl | Pr→Re | Re→Ar | Re→Cl | Re→Pr | Avg. | Hard Avg. |
> |-----------|-------|-------|-------|-------|-------|-------|-------|-------|-------|-------|-------|-------|------|-----------|
> | DivideMix | 51.7  | 74.7  | 78.5  | 61.8  | 72.4  | 73.3  | 59.8  | 48.0  | 82.9  | 68.0  | 56.4  | 81.6  | 67.4 | 58.4      |
> | DINE      | 52.2  | 78.4  | 81.3  | 65.3  | 76.6  | 78.7  | 62.7  | 49.6  | 82.2  | 69.8  | 55.8  | 84.2  | 69.7 | 60.4      |
> | BETA      | 57.2  | 78.5  | 82.1  | 68.0  | 78.6  | 79.7  | 67.5  | 56.0  | 83.0  | 71.9  | 58.9  | 84.2  | 72.1 | 64.4      |
>
> In summary, BETA has significant differences with DivideMix, though being partially inspired by DivideMix. We highlight the technical differences:
>
> 1. We ameliorate the cross-domain knowledge distillation by subdomain division so that the confirmation bias is reduced.
> 2. We propose subdomain alignment to remove internal domain shift.
> 3. We propose subdomain augmentation that is different to DivideMix, inspired by FixMatch.
> 4. We derive the first theorem for the DABP problem, which motivates and explains the design BETA accordingly.

---

> ### Author Response · Authors · 2022-11-14
> **Response to Reviewer zK5K (1/2)**
>
> We sincerely thank the reviewer zK5K for the detailed summary and constructive comments. We are glad that the reviewer acknowledges that the motivation is clear and the problem is novel. Here we answer all the questions and hope they can address the concerns.
>
> **Q1: For a batch of new test data, is it necessary to do extra studies on threshold value to find a suitable value according to different data distributions?**
>
> It is a very interesting question regarding the viability of BETA in the real world. The answer is that it is not necessary to do extra studies on the hyperparameter. As shown in Figure 3(b), for $\tau$ in [0.3, 0.9], the performance of our method only varies from 78.1 to 78.7 on Office-Home, demonstrating that BETA is robust to the hyper-parameter sensitivity. One may argue that Office-Home is not a large dataset to prove this, so we supplement an additional experiment on VisDA-17. The results have been shown as follows:
>
> | Method | plane | bcycl | bus | car | horse | knife | mcycle | person | plant | sktbrd | train | truck | Per-class |
> |---|---|---|---|---|---|---|---|---|---|---|---|---|---|
> | Source-only | 64.3 | 24.6 | 47.9 | 75.3 | 69.6 | 8.5 | 79.0 | 31.6 | 64.4 | 31.0 | 81.4 | 9.2 | 48.9 |
> | DINE | 81.4 | 86.7 | 77.9 | 55.1 | 92.2 | 34.6 | 80.8 | 79.9 | 87.3 | 87.9 | 84.3 | 58.7 | 75.6 |
> | BETA ($\tau$=0.4) | 95.1 | 83.0 | 81.8 | 70.4 | 94.5 | 72.6 | 87.7 | 79.4 | 95.0 | 90.5 | 87.6 | 44.5 | 81.8 |
> | BETA ($\tau$=0.5) | 96.2 | 83.9 | 82.3 | 71.0 | 95.3 | 73.1 | 88.4 | 80.6 | 95.5 | 90.9 | 88.3 | 45.1 | 82.6 |
> | BETA ($\tau$=0.6) | 95.3 | 83.4 | 81.5 | 70.8 | 94.6 | 72.2 | 88.5 | 80.3 | 94.6 | 90.7 | 88.2 | 45.3 | 82.1 |
> | BETA ($\tau$=0.7) | 94.3 | 82.3 | 80.7 | 70.1 | 93.8 | 72.1 | 87.1 | 79.9 | 94.2 | 89.8 | 87.2 | 44.3 | 81.3 |
>
> The results show that with varying hyper-parameters $\tau$, the proposed method can still achieve significant improvements against the source-only model and the existing state-of-the-art method (DINE). Even the worst case ($\tau$=0.4) brings an improvement of 32.9% against the source-only model and outperforms the existing state-of-the-art model (DINE) by 6.2%. Therefore, it is not necessary to do extra studies on the hyper-parameter in real-world applications, and we recommend using the empirical value (0.6$\pm$0.2) that can perform well on all the datasets in this paper. Note that $\tau$ cannot be set to a very large value, as this could lead to a limited number of the easy-to-adapt subdomain. It cannot be set to a very small value, which could lead to a very noisy split for the easy-to-adapt and hard-to-adapt subdomains.

---

> ### Author Response · Authors · 2022-11-18
> **Looking forward to your reply**
>
> Dear reviewer, we have provided new experiments and discussions according to your valuable suggestions, which have been absorbed into our revised manuscript. We hope that the new manuscript is made to be stronger with your suggestions.
>
> As the rebuttal deadline is approaching in one day, we look forward to your reply or new suggestions. Thanks so much!

---

> ### Author Response · Authors · 2022-11-18
> **Feel free to let us know if you have any concerns regarding our response**
>
> Dear reviewer,
>
> We really appreciate your constructive suggestions that make our paper stronger. We have submitted the new manuscript according to your suggestions. As the rebuttal deadline is approaching in several hours, could you please have a look at our response? We are sincerely looking forward to your reply!
>
> Feel free to let us know if you have any other concerns. Thanks!
>
> Best Regards,
>
> Authors of Submission 912

---

### Official Review · Reviewer_JUAJ · 2022-11-01

**Confidence:** 4
**Correctness:** 3
**Technical Novelty And Significance:** 3
**Empirical Novelty And Significance:** 3
**Recommendation:** 8

**Clarity, Quality, Novelty And Reproducibility:**

**Clarity & Quality**
The paper is well written and presented clearly.

**Novelty**
The novelty compared with DivideMix needs to be highlighted.

**Reproducibility**
Good.



**Strength And Weaknesses:**

**Strength**
1. The paper is well motivated and the the considered problem is meaningful.
2. The combination of ideas from LNL and semi-supervised learning are organic, which handle the confirmation bias well.
3. The derived bound helps explain the effectiveness of the proposed method.


**Weakness**
1. The most concerned point is that the proposed framework is very similar to DivideMix [36]. It looks like implementing DivideMix on another topic. The authors need to highlight the novelty compared with DivideMix.
2. Theorem 1 needs to be further explained. It would improve the paper if the conformation bias is proved to be lower than the naive method with the result from Theorem 1.
3. Intuitively, the conformation bias may not always be mitigated by semi-supervised learning (SSL). The benefit of the proposed method may jointly depend on the number of samples in the target domain and the noise rate of hard-domain samples. In algorithms such as MixMatch, it has beed proved that SSL algorithm may hurt some sub-populations if the corresponding pseudo noise rate is high and the unlabeled sample size is small [R1]. It is interesting to see whether there is similar observations in this paper. If so, the conformation bias is not mitigated in this case.
4. Minor: Does $\rho_h$ indicate the noise/error rate of the hard-domain samples?

[R1] Zhu, Z., Luo, T. and Liu, Y., 2021. The rich get richer: Disparate impact of semi-supervised learning. ICLR 2022.

**Summary Of The Paper:**

The paper studies the domain adaptation when the target domain is unlabeled. A black-box predictor trained on the source domain is employed to assign noisy/pseudo labels to the unlabeled target domain data. In this way, the problem is transformed to learning with noisy labels. The noisy label problem is further solved by adapting semi-supervised learning techniques.

**Summary Of The Review:**

The paper considers an interesting problem and proposes an effective solution. Although it seems to be similar to DivideMix, it successfully implement this idea on a different problem. They also have theoretical results to support their method. The paper would be improved if the authors further discuss the relationship between Theorem 1 and the mitigation of conformation bias.

---

> ### Author Response · Authors · 2022-11-13
> **Response to Reviewer JUAJ (2/2)**
>
> **Q2: Theorem 1 needs to be further explained.**
>
> Through theoretical proof, we can show that the target error is mainly bounded by the pseudo label rate of the hard-to-adapt domain $\rho_h$ and the intra-domain discrepancy, which conforms with the design of our learning objectives. Furthermore, we empirically show that $\rho_h$ decreases in the training procedure of BETA. Hence, Theorem 1 can explain our method well, though it truly does not prove that our method has a lower bound of the target error. The detailed proof has been written in the appendix.
>
> **Q3: Intuitively, the conformation bias may not always be mitigated by semi-supervised learning (SSL), such as MixMatch. For BETA, if the corresponding pseudo noise rate is high and the unlabeled sample size is small, is the confirmation bias still mitigated?**
>
> It is a very interesting problem. One example in our experiment is the D$\to$A task on Office-31. The initial noise rate is 56.4% and the target domain only has 2817 images for 31 classes. On average, each class only has less than 100 images. The BETA still achieves 76.1% accuracy with around 20% improvement. Moreover, to further investigate the problem, we add one experiment using Office-Home. We choose four hard tasks Ar$\to$Cl (44.1%), Cl$\to$Ar (54.5%), Pr$\to$Ar (52.8%), Re$\to$Cl (46.7%), and only choose a small subset (, i.e., only 30 samples per class) of the original domain as the unlabeled target domain data. In this case, we further run the source-only model, and our BETA, and compare their results as follows:
>
> | Method      | Ar$\to$Cl | Cl$\to$Ar | Pr$\to$Ar | Re$\to$Cl | Average |
> |-------------|-----------|-----------|-----------|-----------|---------|
> | Source-only | 46.97     | 51.65     | 52.00     | 47.28     | 49.48   |
> | Ours        | 53.79     | 60.32     | 61.10     | 54.26     | 57.37   |
>
> It is seen that even with a high noise rate and a small number of unlabeled samples, our method still achieves a robust improvement of 7.89% on average. This is due to the noise label learning strategy in BETA that helps better overcome the challenge of high noise rate.
>
> Despite the convincing results, we believe that there should exist a critical point as the reviewer points out. When the number of samples is too low with a super high noise rate, the confirmation bias cannot be mitigated well. To this end, we choose a super small subset (, i.e., only 8 samples per class) of the original domain as the unlabeled target domain data, and get the following results.
>
> | Method      | Ar$\to$Cl | Cl$\to$Ar | Pr$\to$Ar | Re$\to$Cl | Average |
> |-------------|-----------|-----------|-----------|-----------|---------|
> | Source-only | 45.77     | 50.96     | 50.96     | 47.50     | 48.80   |
> | BETA        | 40.38     | 43.35     | 45.77     | 40.77     | 42.57   |
>
> The results show that our method cannot work in this setting. As only 8 samples per class are leveraged, this should be considered as a *few-shot domain adaptation* task, which is still a challenging problem that is remained to be tackled in future work.
>
>
> **Q4: Does $\rho_h$ indicate the noise/error rate of the hard-domain samples?**
> Yes. As illustrated in the third row below Eq. (14), $\rho_h$ denotes the pseudo label rate of the hard-to-adapt subdomain.

---

> > ### Comment · Reviewer_JUAJ · 2022-11-18
> > **Thanks for the rebuttal**
> >
> > The new experiments are convincing. Happy to vote for acceptance!

---

> ### Author Response · Authors · 2022-11-13
> **Response to Reviewer JUAJ (1/2)**
>
> We sincerely thank the reviewer JUAJ for the insightful and constructive comments. We are glad that the reviewer acknowledges that the paper is well motivated, the problem is meaningful and the theory helps explain the method. Here we answer all the questions and hope they can address the concerns.
>
> **Q1: What’s the difference between this method with DivideMix? The authors need to highlight the novelty compared with DivideMix.**
>
> We agree with the reviewer, and therefore make the following comparisons with DivideMix. These discussions will be supplemented to the final manuscript. The main differences between BETA and DivideMix are illustrated from four perspectives:
>
> |  | BETA | DivideMix |
> |---|---|---|
> | Task | Domain Adaptation of Black-box Predictior (DABP). the noise of the target domain in DABP is caused by domain shift between two different domains. | Learning with Noisy Labels (LNL). The noise of LNL is randomly added. |
> | Method | (a) BETA uses GMM to divide the target domain and MixMatch with different augmentation for semi-supervised learning. (similar) (b) BETA applies knowledge distillation between models in parallel to the semi-supervised learning, which is purified by the subdomain division to suppress error accumulation during distillation. (c) Subdomain alignment is proposed to align the internal domain shift. (d) Subdomain augmentation is proposed to enhance structural regularization (i.e., mutual information and mix-up). Strong-weak augmentation fully utilizes the high-confidence samples in $X_e$ and single weak augmentation does not introduce more noise to $X_h$. This process enhances the $L_{mi}$ in Eq.(8) which encourages the model to better comply with the cluster assumption and prevents the partiality for categories. | DivideMix uses GMM to divide the data and then use MixMatch for semi-supervised learning. |
> | Theory | BETA analyzes the algorithm design theoretically and its connection with the learning shift of DABP.  A new bound of DABP is derived to explain the rationale behind the optimization. | N.A. |
> | Experiments | We added an experiment on Office-Home to demonstrate the performance difference. As shown in the table below,  BETA outperforms DivideMix by 4.7% on average. For the hard tasks with distant domain shift, BETA outperforms DivideMix by 6.0% on average.  | Experiments are conducted on LNL benchmarks. |
>
> The comparison on Office-Home is shown as follows, and our method outperforms the DivideMix by 6.0% and the state-of-the-art DINE by 4.0% on average.
>
> | Method    | Ar→Cl | Ar→Pr | Ar→Re | Cl→Ar | Cl→Pr | Cl→Re | Pr→Ar | Pr→Cl | Pr→Re | Re→Ar | Re→Cl | Re→Pr | Avg. | Hard Avg. |
> |-----------|-------|-------|-------|-------|-------|-------|-------|-------|-------|-------|-------|-------|------|-----------|
> | DivideMix | 51.7  | 74.7  | 78.5  | 61.8  | 72.4  | 73.3  | 59.8  | 48.0  | 82.9  | 68.0  | 56.4  | 81.6  | 67.4 | 58.4      |
> | DINE      | 52.2  | 78.4  | 81.3  | 65.3  | 76.6  | 78.7  | 62.7  | 49.6  | 82.2  | 69.8  | 55.8  | 84.2  | 69.7 | 60.4      |
> | BETA      | 57.2  | 78.5  | 82.1  | 68.0  | 78.6  | 79.7  | 67.5  | 56.0  | 83.0  | 71.9  | 58.9  | 84.2  | 72.1 | 64.4      |
>
> In summary, BETA has significant differences with DivideMix, though being partially inspired by DivideMix. We highlight the technical differences:
>
> 1. We ameliorate the cross-domain knowledge distillation by subdomain division so that the confirmation bias is reduced.
> 2. We propose subdomain alignment to remove internal domain shift.
> 3. We propose subdomain augmentation that is different to DivideMix, inspired by FixMatch.
> 4. We derive the first theorem for the DABP problem, which motivates and explains the design BETA accordingly.
>
> This work is the first that explores the confirmation bias issue in DABP. Our focus is not to develop brand-new methods. Yet, we aim to develop a simple yet effective solution to deal with confirmation bias and demonstrate that suppressing confirmation bias can revamp the cross-domain knowledge distillation, achieving state-of-the-art performances on all DABP benchmarks.

---

### Official Review · Reviewer_yM8j · 2022-11-02

**Confidence:** 5
**Clarity, Quality, Novelty And Reproducibility:** The novelty of the proposed method is…
**Correctness:** 2
**Technical Novelty And Significance:** 2
**Empirical Novelty And Significance:** 2
**Recommendation:** 5

**Strength And Weaknesses:**

Strengths:

1. The experimental results on some DABP tasks are better than the mentioned works.

Weaknesses:

1. The current paper lacks some necessary references.
2. The proposed method is not novel in domain adaptation field.


**Summary Of The Paper:**

This work studies domain adaptation of black-box predictors (DABP) where the adaptation procedure on target domain has no access to source model and data. To solve this task, this paper proposes a BETA method which divides target samples into easy-to-adapt instances and hard-to-adapt ones and utilizes augmentation manner to obtain more clean samples for model training. This work achieves better performance than the mentioned baselines.

**Summary Of The Review:**

This work studies domain adaptation of black-box predictors (DABP) where the adaptation procedure on target domain has no access to source model and data. To solve this task, this paper proposes a BETA method which divides target samples into easy-to-adapt instances and hard-to-adapt ones and utilizes augmentation manner to obtain more clean samples for model training. This work achieves better performance than the mentioned baselines.

For the current version, I have several concerns.

1. The abstract claims that “an observation we make for the first time … target domain usually contains easy-to-adapt and hard-to-adapt samples…”. However, this observation has been well-observed and studied by many domain adaptation works [1, 2, 3, 4]. These works typically use the probability output of model to determine the easy or hard samples. Thus, this concept is not novel, and the sample division strategy used in Eq. (3) and Eq. (4) is basically same with the existing works.

2. The proposed method is not novel for domain adaptation field. Using the linear combination of samples to augment the high-confident instances has been explored by [5]. The mutual information maximization is also used by many domain adaptation works such as SHOT [6]. Thus, the current method seems to be the combination of many existing methods. In addition, [5] and other recent UDA works have achieved higher performance than the mentioned baselines. However, these works are missing in Table 1-3.

3. When dividing target samples into two sub-domains, the error is naturally divided into two parts to form the inequality as Eq. (14). Thus, Theorem 1 does not illustrate that the proposed method can achieve better sub-domain alignment.

[1] Cui, Shuhao, et al. "Gradually vanishing bridge for adversarial domain adaptation." Proceedings of the IEEE/CVF conference on computer vision and pattern recognition. 2020.

[2] Shu, Yang, et al. "Transferable curriculum for weakly-supervised domain adaptation." Proceedings of the AAAI Conference on Artificial Intelligence. Vol. 33. No. 01. 2019.

[3] Shin, Inkyu, et al. "Two-phase pseudo label densification for self-training based domain adaptation." European conference on computer vision. Springer, Cham, 2020.

[4] Chen, Chaoqi, et al. "Progressive feature alignment for unsupervised domain adaptation." Proceedings of the IEEE/CVF conference on computer vision and pattern recognition. 2019.

[5] Na, Jaemin, et al. "Fixbi: Bridging domain spaces for unsupervised domain adaptation." Proceedings of the IEEE/CVF Conference on Computer Vision and Pattern Recognition. 2021.

[6] Liang, Jian, Dapeng Hu, and Jiashi Feng. "Do we really need to access the source data? source hypothesis transfer for unsupervised domain adaptation." International Conference on Machine Learning. PMLR, 2020.

---

> ### Author Response · Authors · 2022-11-09
> **Response to Reviewer yM8j (2/2)**
>
> *Q4: When dividing target samples into two sub-domains, the error is naturally divided into two parts to form the inequality as Eq. (14). Thus, Theorem 1 does not illustrate that the proposed method can achieve better sub-domain alignment.*
>
> **Answer**: It is true that the error is naturally divided into two parts for two subdomains. However, through theoretical proof, we can show that the target error is mainly bounded by the pseudo label rate of the hard-to-adapt domain $\rho_h$ and the intra-domain discrepancy, which conforms with the design of our learning objectives. Furthermore, we empirically show that $\rho_h$ decreases in the training procedure of BETA. Hence, Theorem 1 can explain our method well, though it truly does not prove that our method has a lower bound of the target error.

---

> ### Author Response · Authors · 2022-11-09
> **Response to Reviewer yM8j (1/2)**
>
> We appreciate the comments from the reviewer yM8j. It is thought that the reviewer may misunderstand some of the illustrations in our paper. We have answered all the questions and sincerely hope they can address the concerns.
>
> *Q1: The observation of easy and hard subdomains has been well-observed and studied by many domain adaptation works [1, 2, 3, 4]. The sample division strategy used in Eq. (3) and Eq. (4) is basically the same as the existing works.*
>
> **Answer**: We would like to point out that there are some misunderstandings in the comment raised by the reviewer:
> 1. Please note that the observation that has been made in our paper for the first time is “the target domain usually contains easy-to-adapt and hard-to-adapt samples that have different levels of domain discrepancy w.r.t. the source domain, and **deep models tend to fit easy-to-adapt samples first**”. The “easy and hard samples” are background, and we use italic font to emphasize the observation "*tend to fit easy-to-adapt samples*" in the abstract. Based on the observation, we warm up the model for several epochs and use a GMM to fit the loss distribution via the EM algorithm. Note that previous methods rely on the sample confidence, i.e., softmax probabilities, for the pseudo labeling, while we model the loss distribution after a warm-up and then perform thresholding based on the loss distribution. Thus, Eq. (3) and Eq. (4) are totally different from the existing works. **As far as we know, we are the first to observe “deep models tend to fit easy-to-adapt subdomain first” in domain adaptation settings and the proposed sample division strategy is mathematically different from the existing pseudo label-based UDA methods.**
> 2. As the reviewer listed, many papers leveraged the pseudo labels or easy-to-hard strategy, but all of these papers [1-4] require the source domain data for training. **Nevertheless, we study the topic of domain adaptation from black-box predictors that do not allow the privacy-preserving UDA algorithm to access the source domain data or models, and thus these papers cannot be used in our scenario.** Specifically, [1] proposes a novel gradually vanishing bridge to achieve a more balanced adversarial UDA based on both source and target domains. [2] proposes to deal with weakly-supervised domain adaptation via curriculum learning, which requires some labeled samples. [3] proposes to enlarge the number of training samples for self-training but still requires the source domain. [4] proposes an easy-to-hard strategy that highly relies on the source domain to accomplish adversarial training w.r.t the easy samples. We will discuss these works in the related work.
>
> *Q2: Why not compare with [5] and other recent UDA methods?*
>
> **Answer**: It is unfair to compare BETA with the standard domain adaptation methods [5], as they can access the source-domain data and model parameters while our method cannot. Without the source-domain data and model parameters, the difficulty is increased significantly. For domain adaptation from black-box predictors, so far the state-of-the-art method is DINE [6] (CVPR’22), which has been compared in all benchmarks and our method outperforms it by a large margin, e.g., 7% average improvement on VisDA-17. In the result table, we list some standard UDA methods only to show that BETA achieves competitive results even when it is compared to standard UDA methods, but it is not necessary to compare BETA with them. As far as we know, BETA achieves state-of-the-art performances on all benchmarks in black-box domain adaptation settings.
>
> *Q3: The method is not novel as Mixup has been used in [5] and mutual information maximization has been used in [6].*
>
> **Answer**: Note that the main contributions lie in the new observation, domain division, and mutually-distilled twin networks. All the other existing techniques are described in Section 3.4 “Algorithmic Instantiation” using only several sentences, as they are not the contribution but are necessary for a strong baseline.
>
> [1] Gradually vanishing bridge for adversarial domain adaptation. CVPR-20
>
> [2] Transferable curriculum for weakly-supervised domain adaptation. AAAI-19
>
> [3] Two-phase pseudo label densification for self-training based domain adaptation. ECCV-2020.
>
> [4] Progressive feature alignment for unsupervised domain adaptation. CVPR-19
>
> [5] Na, Jaemin, et al. "Fixbi: Bridging domain spaces for unsupervised domain adaptation. CVPR-21.
>
> [6] Do we really need to access the source data? source hypothesis transfer for unsupervised domain adaptation. ICML-20
>
> [7] Dine: Domain adaptation from single and multiple black-box predictors. CVPR-22

---

> ### Author Response · Authors · 2022-11-18
> **Looking forward to your reply**
>
> Dear reviewer, we have provided new experiments and discussions according to your suggestions. The comparison between existing works and ours has been made, which demonstrates that our observation is new in the domain adaptation field. More discussions could be found in the following responses.
>
> As the rebuttal period is going to end, may I know if you have any other concerns regarding our work? Thanks!

---

> ### Author Response · Authors · 2022-11-18
> **Feel free to let us know if you have any other concerns**
>
> Dear reviewer,
>
> As the rebuttal deadline is approaching in several hours, could you please have a look at our response? We are looking forward to your reply!
>
> Feel free to let us know if you have any other concerns. Thanks!
>
> Best Regards,
>
> Authors of Submission 912

---

### Author Response · Authors · 2022-11-17
**Summary of major changes in the revision**

We sincerely thank all the reviewers for the helpful comments and suggestions. We now have uploaded the rebuttal version of our paper together with the appendix where the revisions are marked in magenta. Due to the space limit, we put additional experiments in the appendix.

Here is the summary of the major changes we made in the revision:
> 1. We include more references to the related work, and make a comparison w.r.t these works (DivideMix, IntroDA, AdaMatch, easy-hard strategy) and BETA in the appendix.
> 2. We include additional extensive experiments and ablation studies on VisDA-17 to demonstrate the robustness of our method.
> 3. We provide additional experiments using challenging settings to have a better understanding of our models including the limitations.
> 4. We provide the standard deviation for the results.
> 5. We include more discussions about the results and improvement.
> 6. We update some results since some results are achieved to a higher state-of-the-art accuracy with hyper-parameter tuning.
> 7. We shed light on some potential applications of BETA, such as semi-supervised domain adaptation, partial domain adaptation, and multi-source domain adaptation, with an experiment for partial domain adaptation scenario.

---

### Decision · Program_Chairs · 2023-01-20

**Decision:**

Accept: notable-top-25%

**Justification For Why Not Higher Score:**

This paper experienced several criticisms in the first round of reviews, to which authors replied properly but, to my opinion, some of the concerns have not fully fixed. The work was considered enough original for ICLR, but still it was inspired by a set of methods in the literature, so it cannot be considered as a brand new method. The provided extensive experimental analysis and the theoretical study well support the method, hence, I think that spotlight could be a good compromise.

**Justification For Why Not Lower Score:**

Just one reviewer provided a rating below threshold, but I am under-weighting his comments not actually for the complaints highlighted by the authors, but rather for the fact that he did not react to the rebuttal until the very end, and he just raised hs rating from 3 to 5 without writing any work for justification. So doing, we have 4 ratings above threshold (6, 6, 8, 8), which justify a spotlight.

**Metareview: Summary, Strengths And Weaknesses:**

This paper received originally contrasting reviews, almost equally subdivided in above and below threshold. Authors feedback was extensively provided and all the ratings increased above threshold, except one that was anyway raised from 3 to 5, still slightly below threshold.
The main issues addressed by the reviewers regarded the low novelty, and the weak experimental validation, especially in terms of missing comparisons with state-of-the-art methods. Conversely, the paper was appreciated for clearly motivate the work, the importance of the problem addressed (black-box DA, i.e., source-free and source model-free setting), for the accurate experimental analysis (but several issues were also raised in this respect), and for providing a theoretical bound.
Authors have replied extensively to all reviewers' concerns, and discussed about the previous papers - quoted by the reviewers - related to the current work that were not addressed in the original version, and also performed an additional thorough experimental validation and ablation analysis, which has allowed to better figure out the limitations of the proposed method and the best scenarios where it can be applied.
This rebuttal convinced almost all reviewers (and the AC) of the validity of the work, leading its acceptance to ICLR23.


**Note From Pc:**

if the above contains the word "oral" or "spotlight" please see: "oral" presentation means -> notable-top-5% and "spotlight" means -> notable-top-25%. As stated in our emails, we are disassociating presentation type from AC recommendations

**Summary Of Ac-Reviewer Meeting:**

N/A